# Interrogating site dependent kinetics over SiO₂-supported Pt nanoparticles

**Taek-Seung Kim** [1], **Christopher R. O'Connor** [1] **& Christian Reece** [1] ✉

A detailed knowledge of reaction kinetics is key to the development of new more efficient heterogeneous catalytic processes. However, the ability to resolve site dependent kinetics has been largely limited to surface science experiments on model systems. Herein, we can bypass the pressure, materials, and temperature gaps, resolving and quantifying two distinct pathways for CO oxidation over SiO₂-supported 2 nm Pt nanoparticles using transient pressure pulse experiments. We find that the pathway distribution directly correlates with the distribution of well-coordinated (e.g., terrace) and under-coordinated (e.g., edge, vertex) CO adsorption sites on the 2 nm Pt nanoparticles as measured by in situ DRIFTS. We conclude that well-coordinated sites follow classic Langmuir-Hinshelwood kinetics, but under-coordinated sites follow non-standard kinetics with CO oxidation being barrierless but conversely also slow. This fundamental method of kinetic site deconvolution is broadly applicable to other catalytic systems, affording bridging of the complexity gap in heterogeneous catalysis.

Heterogeneous catalytic processes are the foundation of the chemical industry. However, our ability to rationalise and predict the behaviour of these complex industrial catalytic reactions has been largely limited due to the significant complexity gap[1] that exists between our fundamental understanding of catalytic systems and their application. Surface science experiments over planar model catalysts have been able to precisely resolve intrinsic catalytic kinetics and dynamic catalytic behaviour[2–6]; however, their application to real-world catalytic systems is often limited due to perceived pressure, material and temperature gaps[7]. These gaps are even more apparent in small (≲5 nm) supported nanoparticle systems where the metals no longer retain their bulk-like properties and notable support-metal interactions can exist[8,9]. Therefore, a method of directly measuring intrinsic kinetics over complex multi-faceted supported nanoparticle catalysts is highly desirable.

As the rate of a reaction is defined by the specific geometric and electronic characteristics of an active site, we propose that measurement of an intrinsic rate constant (directly related to the free energy of the reaction) would be specific to a given active site. Therefore, a precisely resolved kinetic coefficient (or set of kinetic coefficients) can act as a parameterised representation of a given active site. CO oxidation has been utilised as a fundamental model reaction for understanding surface catalytic processes for decades[4,10–13]. In particular, CO oxidation over Pt catalysts has attracted significant attention due to its complex and dynamic surface chemistry, with the reaction generally understood to proceed via a Langmuir-Hinshelwood mechanism[14,15]. More recently, CO oxidation was used as a probe to study atom utilisation from nanoparticles to the single-site limit[13,16]. Due to the variations in physical structure with changing nanoparticle size[17] and reaction environment[18,19] understanding the catalytic role of geometric surface sites (i.e., structural-sensitivity) has been[20,21], and still is[22] of great interest. For example, the activity of Pt catalysts has been considered size-dependent for C–H bond activation in the methane reforming reaction[23,24], but conversely their activity for CO oxidation has typically been regarded as size-independent[15]. However, the possibility of site-dependence for CO oxidation on non-reducible oxide-supported Pt catalysts was recently reported[25,26] where through a combination of *operando* DRIFTS and steady-state kinetic measurements it was thought that different reaction kinetics for under-coordinated (UC) and well-coordinated (WC) sites exists on the metallic Pt surface. However, as the study was performed under steady state conditions their analysis was limited to simple reaction orders and apparent activation energies.

Non-steady state techniques such as Temporal Analysis of Products (TAP, see section: The Temporal Analysis of Products

[1]Rowland Institute at Harvard, Harvard University, Cambridge, MA 02142, USA. ✉e-mail: christianreece@fas.harvard.edu

Experiment) are an effective way of measuring intrinsic kinetics as they can provide information about each sequential elementary step for the overall reaction. The TAP experiment serves to bridge the perceived pressure, material, and temperature gaps with the peak pressure during a pulse over the catalyst being on the order of 1 mbar[27] (similar to other in-situ methods) and by using a packed bed microreactor (allowing powdered samples) which can be heated to the reaction temperature. As the pulse of reactant gas contains significantly fewer molecules than the number of reactive sites on the catalyst, a single pulse is not considered to change the surface significantly. However, by repeatedly pulsing it becomes possible to dynamically evolve the catalyst state using a technique known as *chemical calculus*[28] making titration-like experiments extremely powerful[29–33].

In this work we demonstrate that with the combination of TAP experiments, kinetic modelling, and in-situ DRIFTS measurements it is possible to precisely resolve site-dependent kinetics on working catalysts. We provide a detailed insight into the site-dependent intrinsic kinetics for CO oxidation over a well-defined 2 nm-sized Pt/SiO₂ catalyst. Using isotopic labelling, we deconvolute the production of $CO_2$ that arises from the reaction with preadsorbed $^{13}CO^*$ and from the adsorption/reaction of reactant $^{12}CO$ in the gas phase under CO oxidation conditions. We identify two distinct kinetic features in the adsorbed $^{13}CO^*$-driven $CO_2$ production, which when combined with isothermal titration experiments are identified as two distinct pathways for CO oxidation. Regression of a kinetic model to the TAP exit flux curves was used to quantify the distribution of each pathway and calculate the intrinsic kinetics for the surface reaction of adsorbed oxygen with preadsorbed $CO^*$ as a function of temperature and coverage. Finally, by combining this data with comparable DRIFTS measurements, we are able to deduce site-specific kinetics as a direct relationship between the distribution of kinetic pathways and the distribution of well-coordinated (e.g., terrace like) and under-coordinated (e.g., edge, vertex) sites.

## Results and discussion

### Initial characterisation of pristine and $CO^*$-covered 2 nm-sized Pt/SiO₂ catalysts

Uniform and well-dispersed Pt nanoparticles were synthesized via a conventional polyol method (Supplementary Fig. 1) and were deposited on a SiO₂ support with a concentration of 0.72 wt%. The average size and distribution of the nanoparticles was measured to be 1.82 ± 0.51 nm (Fig. 1a) on the fresh catalyst, and 1.94 ± 0.37 nm (Fig. 1b) over the spent catalyst. As no sintering of the nanoparticles occurred throughout the duration of the TAP experiments, we can directly correlate the catalytic behaviour to this narrow distribution of particles approximately 2 nm in diameter. The catalytic activity of the 2 nm Pt/SiO₂ catalyst was tested under steady-state CO oxidation conditions (2.5% CO, 5% O₂) with an apparent activation energy of 89 ± 3 kJ/mol measured between 130 and 160 °C, and reaction orders of − 1.02 ± 0.06 in CO and 1.22 ± 0.1 in O₂ (Supplementary Fig. 2). These

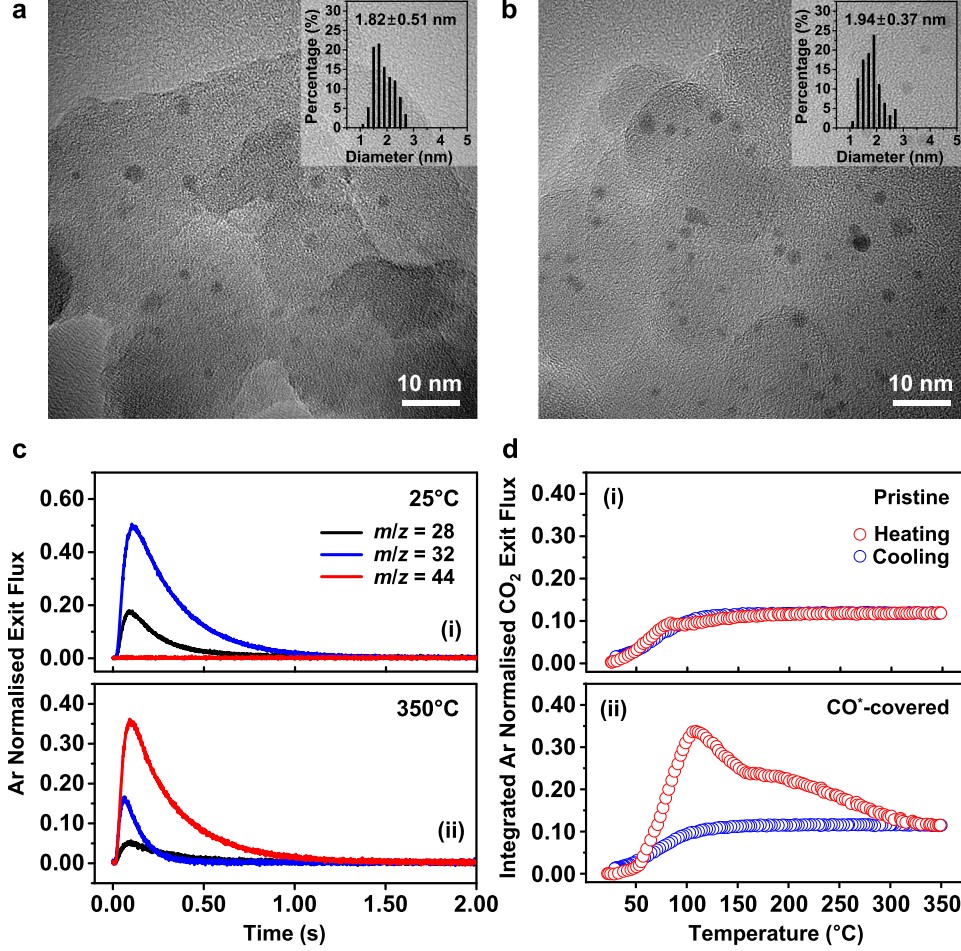

**Fig. 1 | Characterisation of Pt/SiO₂ catalyst and TAP CO oxidation experiments.** **a, b** Representative TEM images with particle size distribution (inset) of (**a**) fresh and (**b**) spent Pt/SiO₂ catalyst after all of the TAP experiments. **c** Argon normalised exit flux curves of m/z = 28 (CO), 32 (O₂), and 44 (CO₂) at (i) 25 and (ii) 350 °C for a pulse set of 6.6% CO 13.4% O₂ gas mixture in an inert Ar tracer over CO*-covered Pt/SiO₂ catalyst. **d** Temperature-dependent integrated exit flux of m/z = 44 (CO₂) normalised via Ar on (i) pristine and (ii) CO-covered Pt/SiO₂ catalyst for CO oxidation while heating from 25–350–25 °C at a heating rate of 8 °C/min.

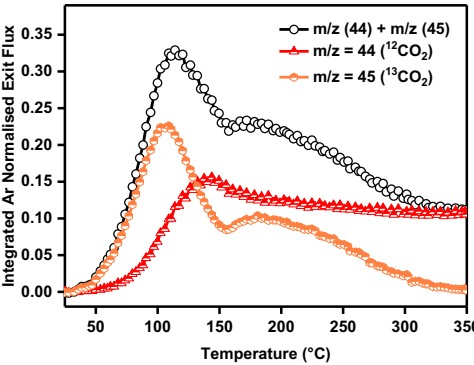

**Fig. 2 | Isotopic labelling TAP experiment.** Temperature-dependent integrated exit flux of m/z = 44 ($CO_2$) and m/z = 45 ($^{13}CO_2$) normalised via Ar from the TAP experiment where an oxygen rich $CO/O_2$ mixture (1:2 molar ratio, 6.6% CO 13.4% $O_2$) was pulsed over a $^{13}CO^*$-covered $Pt/SiO_2$ catalyst while heating from 25–350 °C at a heating rate of 8 °C/min.

values match well with previous results over Pt catalysts with similar average particle sizes[15,25,34–36], demonstrating that the $Pt/SiO_2$ catalyst used in this work is comparable with ones reported previously, albeit with a narrower size distribution. Finally, we see excellent reproducibility for all TAP experiments performed in this work (Supplementary Fig. 3).

The fresh 2 nm $Pt/SiO_2$ catalyst was loaded into a home-built TAP reactor[37] and the catalytic activity for CO oxidation was measured by pulsing an oxygen rich $CO/O_2$ mixture (1:2 molar ratio, 6.6% CO 13.4% $O_2$) over a pristine and $CO^*$-covered catalyst while heating from 25 to 350 °C at a heating rate of 8 °C/min. The 1:2 ratio of $CO:O_2$ was chosen to allow excess $O_2$ in the reactant stream to act as a titrant to sequentially remove the preadsorbed $CO^*$ to probe the catalytically activity as a function of $CO^*$ coverage. At 25 °C no conversion of CO or $O_2$ and no production of $CO_2$ was measured (Fig. 1c-(i)), whereas at 350 °C near 100% conversion of the CO to $CO_2$ is recorded (Fig. 1c-(ii)). To simplify the comparison between the experiments, the exit flux curves for every pulse set in the experiment were integrated and normalised to the inert Ar tracer (Fig. 1d). The pristine $Pt/SiO_2$ catalyst (Fig. 1d-(i)) shows a gradual increase in activity from 0% CO conversion at room temperature to almost 100% CO conversion above 100 °C. Similar reactivity for $CO_2$ production between the heating and cooling steps was found indicating the catalyst state remains consistent during the experiment. However, the $CO^*$-covered catalyst (Fig. 1d-(ii)) shows increased production of $CO_2$ compared with the pristine catalyst. The excess $O_2$ in the reactant gas mixture is able to react with the preadsorbed $CO^*$ and act as a titrant, sequentially removing the preadsorbed reactive sites with increasing temperatures. By 350 °C it is assumed that almost all of the preadsorbed $CO^*$ has either been reacted or desorbed off the catalyst surface as the activity during cooling was the same as that measured over the pristine catalyst. Interestingly, during the heating ramp the $CO^*$-covered 2 nm $Pt/SiO_2$ catalyst showed two catalytic features, with peaks in $CO_2$ production around 100 and 200 °C. However, further experiments were required to precisely understand the cause for these two kinetic features.

**Identifying pathways for oxidation of preadsorbed $CO^*$**

In an attempt to further rationalise the two kinetic features of observed in Fig. 1d-(ii), isotopically labelled $^{13}CO$ was used to prepare a $^{13}CO^*$-covered catalyst. This allows a precise deconvolution of the $CO_2$ produced from the reaction of the 6.6% CO 13.4% $O_2$ gas mixture over the catalyst (m/z = 44) and the $CO_2$ produced from the reaction of $O_2$ with preadsorbed $^{13}CO^*$ (m/z = 45). The heating rate experiment was repeated on the $^{13}CO$-covered $Pt/SiO_2$ catalyst under the exact same conditions as used in Fig. 1d-(ii). Interestingly, the reaction of gas phase CO

and $O_2$ with the catalyst (Fig. 2, red triangles) was similar to that measured over the pristine catalyst whereas the preadsorbed $^{13}CO^*$-driven $CO_2$ production (Fig. 2, orange circles) shows two well-defined kinetic features around 100 and 200 °C. The total $CO_2$ production ($^{12}CO_2 + {}^{13}CO_2$, Fig. 2, black circles) also matched well with the $CO_2$ measured in Fig. 1d-(ii). As in other temperature programmed techniques such as Temporal Programmed Oxidation (TPO), the two peaks in the measured signal would indicate two different kinetic pathways (often prescribed to different sites) for the oxidation of preadsorbed CO with gas phase $O_2$.

To directly probe the kinetic features at 100 and 200 °C, isothermal $O_2$ titration experiments were carried out. First, CO was pulsed over the catalyst at the reaction temperature until it was saturated with adsorbed $CO^*$. Then, the preadsorbed $CO^*$ was titrated off sequentially with a series of $O_2$ pulses. This affords calculation of the apparent rate constant for the oxidation of $CO^*$ as a function of CO coverage[30,33]. In the first few pulses ~100% conversion of $O_2$ is observed at both temperatures (Fig. 3-left). Then, as $O_2$ is repeatedly pulsed, the $CO_2$ production decreases, and the exit flux of $O_2$ increases up to the saturation state on the Pt surface, confirming that complete removal of the reactive $CO^*$ occurs. To estimate the kinetics of the preadsorbed $CO^*$ consumption, we plot the temperature corrected apparent rate constant as a function of cumulative $CO_2$ produced (indicative of $CO^*$ coverage) in Fig. 3a, b-right, which is calculated using[30]:

$$k'_{app} = \frac{X}{1-X}\sqrt{T} \approx k_{app} \approx k_a \theta_{CO^*} \tag{1}$$

Where $k'_{app}$ is the temperature corrected apparent rate constant, $X$ is the fractional $O_2$ conversion, $T$ is the temperature (K), $k_{app}$ is the apparent rate constant (s$^{-1}$), $k_a$ is the intrinsic rate constant for the reaction of $O_2$ with preadsorbed $CO^*$, and $\theta_{CO^*}$ is the coverage of preadsorbed $CO^*$. On account of the usage of the same catalyst bed throughout all the pulse experiments, variations in $k_{app}$ show a direct correlation to the intrinsic rate constant for the $O_2$ titration experiment[38,39]. Specifically, a linearly decrease in $k_{app}$ with $\theta_{CO^*}$ (or $CO_2$ produced) indicates the presence of a unique intrinsic rate constants ($k_a$) for the reaction of $O_2$ with $CO^*$ through the following relationship[30,40]:

$$\frac{\Delta k_{app}}{\Delta \theta_{co^*}} = \frac{(1-\varepsilon)}{\varepsilon V}k_a \tag{2}$$

Where $\varepsilon$ is the void fraction of the packed bed, and V is volume of the catalyst zone. We identify two linear regimes for the reaction of $O_2$ with preadsorbed $CO^*$ in both isothermal titration experiments (Fig. 3, yellow line), indicating that two intrinsic rate constants, and as such two pathways exist simultaneously at both temperatures. We identify a slow and fast intrinsic reaction rate for $CO_2$ production, as shown by the gradients in Fig. 3-right. At high relative coverage of $CO^*$ the fast pathway dominates, at lower $CO^*$ coverages the slow pathway dominates during the titration experiment. Coupling this insight with the two kinetic regimes seen in the temperature programmed experiments, we feel confident in claiming that at least two different pathways for the oxidation of $CO^*$ by $O_2$ exist on the 2 nm $Pt/SiO_2$ catalyst.

**Quantifying active species using MZTRT and DRIFTS**

Due to its well-defined nature, it is possible to precisely resolve the intrinsic kinetics of catalytic processes using the TAP experiment[39,41]. For linear (first order or pseudo first order) reactions Multi-Zone TAP Reactor Theory[42,43] (MZTRT) is a powerful and efficient tool for simulating TAP exit flux responses. The model of the experiment was built using the generalised form of MZTRT (see Supplementary I and Supplementary Dataset 1) with the Symmetric Thin-Zone assumption

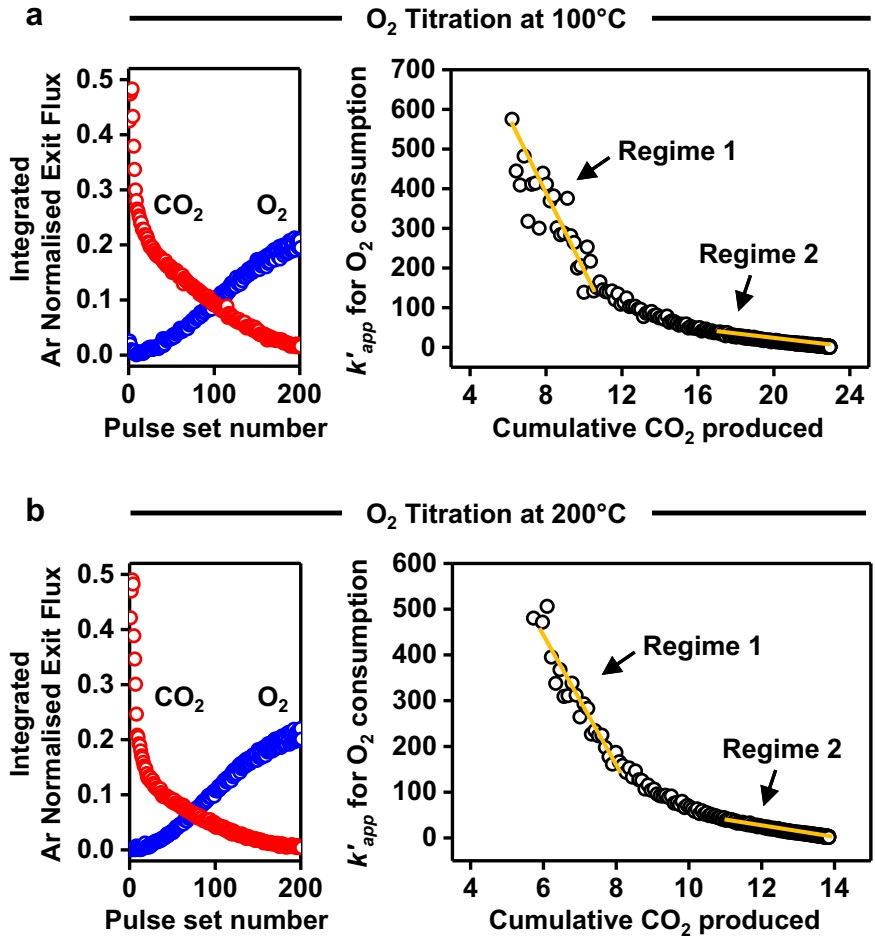

**Fig. 3 | Isothermal O$_2$ titration experiments at 100 and 200 °C over the CO$^*$-covered Pt/SiO$_2$ catalyst. a, b** Integrated Ar normalised exit flux of O$_2$ and CO$_2$ during the O$_2$ titration experiments over the CO$^*$-covered Pt/SiO$_2$ catalyst (left), and the apparent rate constant ($k'_{app}$) as a function of cumulative CO$_2$ produced (right) at (**a**) 100 and (**b**) 200 °C. Yellow lines are provided to guide the eye to the two kinetic regimes.

applied to the catalyst zone[39]. For the reaction of oxygen with preadsorbed CO$^*$, a three-pathway model was identified as the most likely candidate (see Supplementary II; Fig. 4a and Supplementary Fig. 4). The experimental results for two catalytic features (CO$_2$ production around 100 and 200 °C in Fig. 2) and two kinetic regimes (two pathways for the reaction of O$_2$ in Fig. 3) would indicate two separate pathways for oxidation of CO$^*$. As the oxygen balance (i.e., oxygen released as CO$_2$ / oxygen consumed) is not always 1 throughout the experiment a third pathway of irreversible oxygen adsorption is necessary in the model. It is very important to note that the model regression is performed on each set of exit flux curves individually (Fig. 4b) with the model fit to both the shape and magnitude of Ar, O$_2$, and CO$_2$ exit flux curves. The kinetic model can be broken down into two parts. First, the apparent adsorption rate constants $k'_{a,1}$ and $k'_{a,2}$ represent the irreversible adsorption and subsequent reaction of gaseous O$_2$ with preabsorbed CO$^*$, and the adsorption rate constant $k'_{a,3}$ represents the irreversible adsorption of O$_2$ to sites where no further reaction with CO$^*$ takes place. Second, the intrinsic rate constants $k_{r,1}$ and $k_{r,2}$ represent the rate at which the surface reaction between adsorbed oxygen and CO$^*$ occurs for pathways 1 and 2, respectively. In short, the adsorption constants $k'_a$ control the magnitude of the O$_2$ and CO$_2$ exit flux curves, whereas the surface reaction constants $k_r$ control the shape. As each pulse set is regressed individually during the isothermal O$_2$ titration experiment (Fig. 4c), a set of rate constants for the reaction of oxygen with preadsorbed CO$^*$ can be calculated at each point (Fig. 4d). Models of varying complexity were

tested, but only the three-pathway model was able to precisely recreate the experimental data without being overparameterised (see Supplementary II; Supplementary Fig. 5). The model fitting exhibits a high degree of confidence and consistency with all TAP experiments in this work, although decreased confidence in the intrinsic surface reaction constants is observed at the limit of very low CO$_2$ production.

To rationalise if the two pathways for oxidation of the preadsorbed CO$^*$ were correlated to the geometric structure of metallic Pt sites (e.g., terrace, edge, vertex sites) on the 2 nm nanoparticles, a series of DRIFTS and TPO experiments were performed where CO was preadsorbed at 25–350 °C (Fig. 5). In general, it is known that the binding energy for adsorbed CO on the under-coordinated (UC) sites (e.g., edge, vertex sites) is higher than that on the well-coordinated (WC) sites (e.g., terrace sites). This implies that when CO is adsorbed at higher temperatures, an increase in populated UC sites relative to the populated WC sites would be expected[44].

From the DRIFTS investigation in Fig. 5a, it is shown that the preadsorption of CO at 35, 100, 200, and 350 °C (35CO, 100CO, 200CO, 350CO, respectively) populates linear bound CO to WC sites, UC sites, and multi-bound CO as evidenced by three v(C−O) bands (Fig. 5a, b features 1-3). The 35CO yields v(C−O) bands centred at 2075 cm$^{-1}$ with a small high frequency shoulder (feature 1), a small band near 2042 cm$^{-1}$ (feature 2), and a broad band at 1805 cm$^{-1}$ (feature 3). Increasing the adsorption temperature to 100 °C causes a slight decrease in the intensity of feature 1 and a shift in features 2 and 3 to lower frequency. For the 200CO and 350CO, there is a dramatic

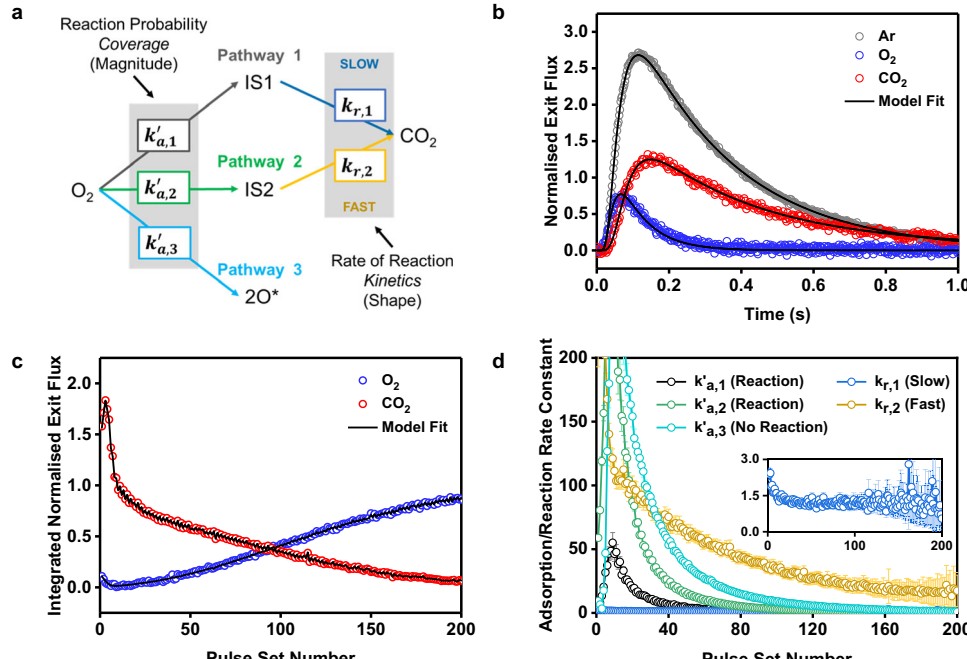

**Fig. 4 | Model fitting of isothermal O₂ titration experiment using MZTRT.**
**a** Schematic of the three-pathway model. **b** Experimentally measured and model exit flux curves for pulse set 40 of O₂/Ar during O₂ titration experiment at 100 °C. **c** Experimentally measured and model fitted integrated exit flux curves for the whole O₂ titration experiment at 100 °C (corresponding to Fig. 3a). **d** Rate constants calculated from model fitting with the error bars representing the 95% confidence intervals for the fitted variables. Inset shows small but non-zero value for $k_{r,1}$.

decrease in the intensity and slight shift to lower frequency of feature 1. The decrease in the intensity is attributed to the reduction of the total number of adsorbed CO* with increasing temperature. In turn the surface is predominantly relatively strongly bound CO on UC sites. A larger shift to lower frequency of features 2 and 3 is observed, which clarifies feature 2 as a distinct peak from feature 1. The position of the v(C − O) frequencies as a function of CO adsorption temperature are reported in Supplementary Table 1. Feature 1 with a peak maximum from 2075 to 2052 cm⁻¹ is assigned to the collective oscillation of linear bound CO to WC sites (linear WC) where the high frequency shoulder is attributed to a dense CO phase for high CO coverages. Feature 2 with a peak maximum from 2042 to 1970 cm⁻¹ is assigned to the collective oscillation of linear CO to UC sites (linear UC). The frequency of the v(C − O) for linear CO on the 2 nm Pt nanoparticles catalyst is in agreement with previous investigations when considering the frequency is dependent on the CO coverage[25,45–55], Pt coordination environment[25,34,49–61], and nanoparticle size[25,55–59]. Feature 3 with a peak maximum from 1805 to 1701 cm⁻¹ is assigned to the collective oscillation of multi-bound CO to Pt sites and is also in reasonable agreement with previous work[47–52,62–66]. In contrast to the linear CO vibrational features, the identification of unique multi-bound CO features for different Pt coordination environments was not possible. The CO* adsorption site information from the DRIFTS analysis is graphically summarised in Fig. 5b.

Alongside the DRIFTS investigation, comparable TPO experiments were performed where CO* was preadsorbed at temperatures from 25 to 350 °C and was titrated using O₂ pulses while heating from 25 to 350 °C at 8 °C/min (Fig. 5c). Interestingly, the 25CO TPO shows an onset temperature for CO₂ production at around 40 °C, whereas the TPO results of both 100CO and 200CO show 10-20 times higher CO₂ production at 40 °C. It is believed that when CO is preadsorbed at 25 °C the surface is fully covered in CO* meaning there are no vacant sites for adsorption/dissociation of O₂ (i.e., CO poisoning). We prescribe the increased activity for CO₂ production in the 100CO and 200CO TPO experiments to the increased number of vacant sites on the partially CO*-covered surface. This is supported by an additional TPO experiment where CO is pulsed to saturation over the 100CO catalyst at 25 °C. The 100CO + 25CO TPO experiment shows similar tendency to that of 25CO (Supplementary Fig. 6), which implies that the low temperature CO₂ production at 25 °C is vacancy driven. Further, isothermal titration experiments performed at 25 °C show no CO₂ production (Supplementary Fig. 7). When combined, this would indicate that CO₂ production by oxidation of preadsorbed CO* is driven by a Langmuir-Hinshelwood type reaction (as is expected for CO oxidation over Pt)[14,15]. However, we cannot rule out other methods such as CO* assisted O₂ dissociation[15]. Interestingly, the low-temperature oxidation of pre-adsorbed CO* would indicate that the Langmuir-Hinshelwood surface reaction between oxygen and CO* over 2 nm Pt nanoparticles has a very low activation barrier, which is counter to previous work on single crystals and supported Pt catalysts with activation energies ranging from 37 to 85 kJ/mol[15,35,67,68].

Similar to the isotopically labelled experiments, two kinetic features are observed: a spike around 80 °C (first feature) and a long shoulder from 100 to 350 °C (second feature) appear in the 25CO TPO. The first kinetic feature appears at low temperatures (< 50 °C) in both the 100CO and 200CO which we ascribe to the low-temperature CO* conversion mentioned above. Due to decreasing CO* coverage throughout the titration experiment, the CO₂ production would be expected to decrease. However, in the 100CO experiment a distinct second catalytic feature appears around 125 °C, which suggests the presence of temperature-dependent kinetics. Further, the second catalytic feature disappears as the adsorption temperature increases from 100 to 200 °C, which corresponds to the large decrease in WC sites from the DRIFTS results in Fig. 5a. This strongly suggests that site dependent kinetics does exist for the oxidation of CO* over SiO₂-supported 2 nm Pt nanoparticles.

To calculate the intrinsic kinetics at each point in the TPO experiments, the rate constants for the model in Fig. 4a were calculated by regressing the MZTRT Symmetric Thin-Zone model to every

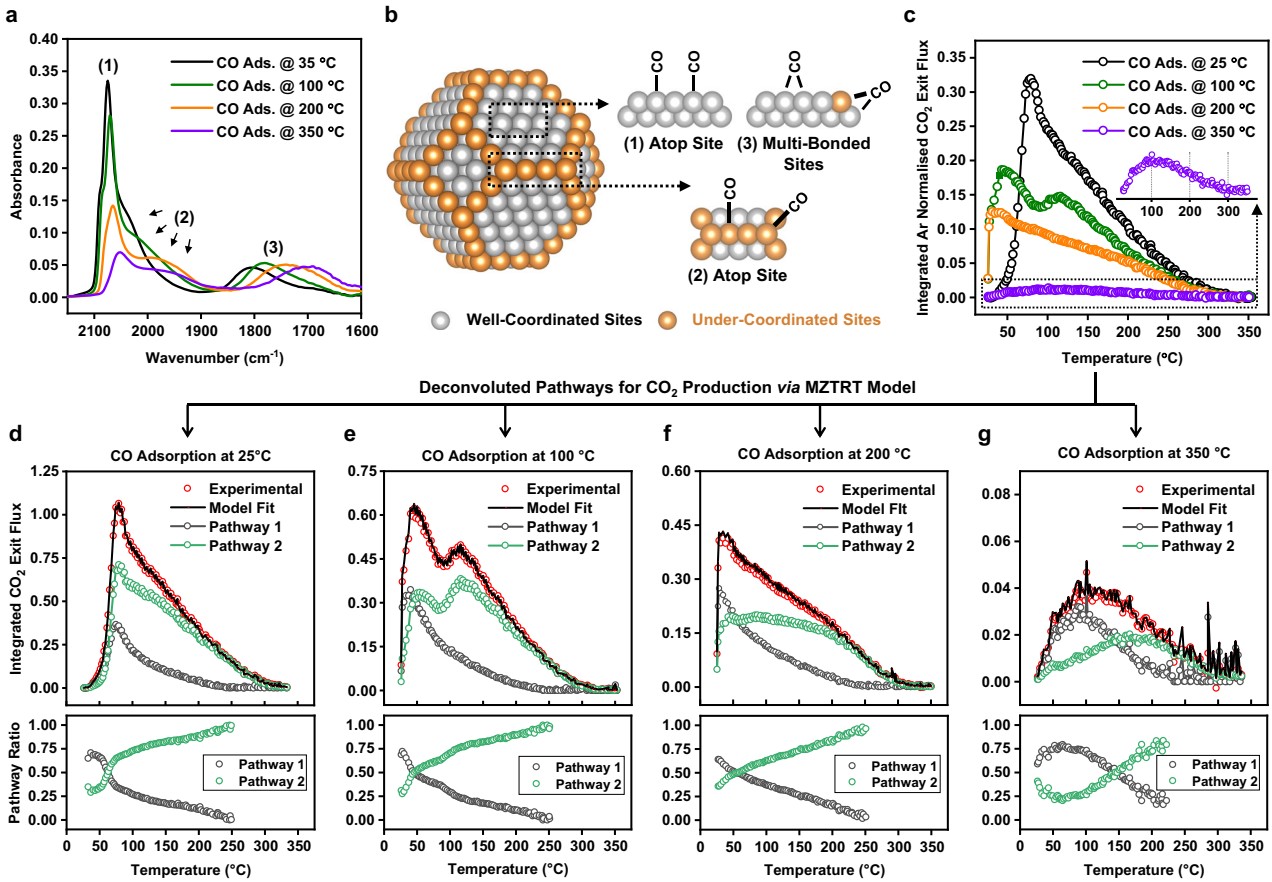

**Fig. 5 | Characterisation of CO* covered catalyst via DRIFTS, TAP, and kinetic modelling. a** DRIFT spectra for CO adsorption on the CO*-covered 2 nm Pt/SiO₂ catalyst where CO* was preadsorbed at 35, 100, 200, and 350 °C. The spectra were obtained after cooling the catalyst to 35 °C. All DRIFTS measurements were acquired at 35 °C which avoids the influence of temperature on vibrational features[80]. **b** Illustration for adsorbed CO* sites (well-coordinated and under-coordinated) on the surface of a model Pt nanoparticle (regular, truncated octahedron; 586 atoms; ~2.17 nm-size). **c** Integrated Ar normalised exit flux of m/z = 44 (CO₂)

during TPO experiments on the CO*-covered Pt/SiO₂ catalyst where CO* was pre-adsorbed at 25, 100, 200, and 350 °C. For TPO experiments, the CO* was pre-adsorbed at 25–350 °C on the catalyst and the catalyst was cooled to 25 °C. Then O₂ was repeatedly pulsed over the catalyst while being linearly heated to 350 °C at 8 °C/min. **d–g** Deconvoluted CO₂ production pathways and CO₂ production pathway ratios calculated using the regressed kinetic model (MZTRT) for the TPO experiments where CO* was preadsorbed at (**d**) 25, (**e**) 100, (**f**) 200, (**g**) 350 °C.

pulse set in the experiments. All rate constants for the 25CO, 100CO, 200CO, and 350CO TPO experiments are shown in Supplementary Fig. 8. Also, the MATLAB script used to process the 25CO TPO experiment is included alongside the Supplementary Information. As during the TAP experiment the pulse size is sufficiently small that coverage of CO* species is not changed by any appreciable amount during a pulse, the first order irreversible adsorption rate constants $k'_{a,1}$ and $k'_{a,2}$ are pseudo first order and are proportional to the concentration of the adsorbed CO* species involved in each pathway ($k'_{a,3}$ would be proportional to the number of empty sites). This means that it becomes possible to deconvolute the amount of CO₂ produced from each pathway as shown in in Fig. 5d–g. From the MZTRT Symmetric Thin-Zone model, the conversion of O₂ in each pulse set can be calculated using[39,69]:

$$X_{O_2} = \frac{\left(k'_{a,1} + k'_{a,2} + k'_{a,3}\right)(L/2D_e)}{1 + \left(k'_{a,1} + k'_{a,2} + k'_{a,3}\right)(L/2D_e)} \quad (3)$$

Where $X_{O_2}$ conversion of the reactant, $D_e^R$ is the diffusivity of the reactant (cm²/cm³), $L$ is the length of the catalyst bed (cm), and $k'_{a,n}$ represents the apparent pseudo-first order adsorption/reaction constant for each individual site included in the model (cm s⁻¹) and is the parameter that is calculated during the model fitting. The apparent pseudo-first order adsorption/reaction constant is linearly related to

the intrinsic adsorption/reaction rate constant through the following relationship[40,42]:

$$k'_{a,n} = k_{a,n}\theta_n L_{cat} \frac{S_v(1 - \varepsilon_b)}{\varepsilon_b} \quad (4)$$

Where $k_{a,n}$ is the intrinsic adsorption/reaction constant (cm³ mols⁻¹ s⁻¹) and $\theta_n$ is the concentration of CO* in the case of sites 1 and 2, and the coverage of empty irreversible adsorption sites in the case of site 3 (mol cm⁻³), $L_{cat}$ is the length of the catalyst zone (cm) $S_v$ is the surface area of the catalyst per volume of catalyst (cm²/cm³). This is a slight modification to forms of the equation previously published[39], as the MZTRT Symmetric Thin-Zone model does not explicitly include a value for $L_{cat}$ and so it is lumped into the apparent pseudo-first order adsorption/reaction constant $k'_{a,n}$. For each pathway included in the model, the conversion of oxygen specific to each pathway can be deconvoluted using the following:

$$X_{O_2,n} = \frac{k'_{a,n}(L/2D_e)}{1 + \left(k'_{a,1} + k'_{a,2} + k'_{a,3}\right)(L/2D_e)} \quad (5)$$

The amount of CO₂ produced through can be calculated from the conversion of oxygen specific to pathways 1 and 2 by considering the

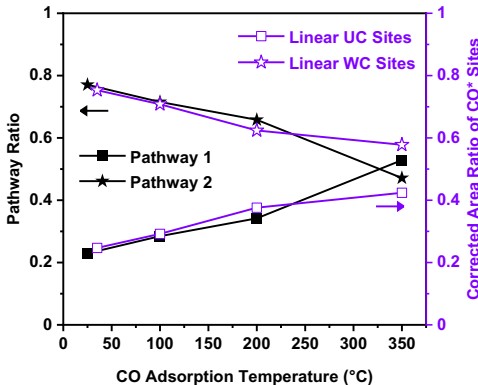

**Fig. 6 | Comparison between pathway ratio (TPO experiment) and area ratio of CO\* sites (DRIFTS investigation).** A direct correlation between the total amount of pathway 1 and pathway 2 calculated from the kinetic model, and the total amount of UC and WC sites from the DRIFTS investigation is found.

reaction stoichiometry using the following:

$$M_{0,norm}^{CO_2,n} = 2X_{O_2,n} \qquad (6)$$

From the pathway deconvolution in Fig. 5d–g, we find that pathway 1 is dominant for $CO_2$ production at low-temperatures, whereas at elevated temperatures pathway 2 becomes the dominant pathway. The $CO_2$ yield from each pathway can be easily calculated by summing the deconvoluted $CO_2$ production throughout the TPO experiment (Supplementary Fig. 9). We find that the $CO_2$ yield of pathway 1 stays relatively constant up to a CO adsorption temperature of 200 °C, whereas the $CO_2$ yield of pathway 2 rapidly decreases. Further, when CO is preadsorbed at 350 °C, the amount of CO species related to pathways 1 and 2 becomes approximately equal on the surface.

To identify if any correlation between the two pathways and geometric surface sites exists, we summarised the ratio of the $CO_2$ yield of pathway 1 to pathway 2 calculated from the kinetic model with the quantitative DRIFTS analysis for the population of linear UC and WC CO\* sites in Fig. 6. The fraction of pathway 1 from the kinetic model (Fig. 6, black) increases with increasing CO adsorption temperature. From the quantitative deconvolution of the DRIFTS spectra for the population of UC and WC CO\* sites (see Supplementary III for details on quantification), the fraction of populated linear CO at UC sites is also increasing with CO adsorption temperature. Deconvolution of the multi-bound CO to WC and UC sites is not possible, but there is a shift to lower frequency of the multi-bound peak maximum with temperature that is consistent with increasing the relative population of UC Pt sites. Most notably, we find a straightforward relationship between the amount of each pathway as calculated from the kinetic model, and the total amount of UC and WC sites calculated using DRIFTS. Due to this direct relationship, we claim that pathway 1 mainly occurs on UC sites, and pathway 2 occurs mainly at WC sites.

### Measuring the Intrinsic Kinetics of CO oxidation on UC and WC sites

Along with the reactive site information, the intrinsic surface reaction constants $k_{r,1}$ and $k_{r,2}$ (related to the reaction of adsorbed oxygen with adsorbed CO for pathways 1 and 2) from the isothermal titration and TPO experiments were calculated and are shown in Fig. 7. As mentioned previously, the reaction over UC sites is ascribed to pathway 1, whereas the reaction over WC sites is ascribed to pathway 2. It can be seen that the rate constant for pathway 1 ($k_{r,1}$) is significantly lower than pathway 2 ($k_{r,2}$) under all conditions probed, with pathway 1 being a slow reaction between adsorbed oxygen and CO and pathway 2 being fast. Interestingly, pathway 1 shows no dependence on temperature or the

coverage of CO in both the isothermal $O_2$ titration (Fig. 7a, b) and the TPO (Fig. 7c, d) experiments but is unintuitively a slow reaction. While it is now possible to identify where the reaction is occurring, the intrinsic rate constants alone cannot provide specifics on how the reaction is occurring without further characterisation methods. However, one such rationalisation for this behaviour based on transition state theory would be that the reaction results in a significant loss in entropy in the transition state[70]. As adsorbed oxygen is considered to be immobile under these conditions, it could be hypothesized that a CO molecule strongly bound to a vertex of a nanoparticle has a large number of degrees of freedom, but during the transition state to make $CO_2$ the species would become immobile due to the strongly bound oxygen, losing a large number of degrees of freedom. However, as CO adsorption on Pt is not well described by DFT[71,72], this would be difficult to confirm, so it is important to note that this idea remains conceptual. Experiments using molecular beam scattering[14,67] or velocity resolved kinetics[73] do not observe this barrierless reaction, but they were performed on Pt single crystals, whereas this work is performed on 2 nm Pt supported nanoparticles, and as such the nanoparticle size effect must also be considered. Other experiments have reported that the barrierless (or near barrierless) oxidation of CO can occur via a $Pt(CO_3)$ intermediates[74], or through an Eley-Rideal type mechanism[15]. While we do not find any direct evidence of those pathways, it is also not possible to entirely rule them out. Further, due to the slow nature of the reaction, this pathway would be difficult to isolate as it would certainly be limited by CO desorption under steady-state conditions at any appreciable CO pressure.

We find that pathway 2 shows more classic kinetic behaviour. Under isothermal conditions we find a linear dependence on the coverage of CO\* (Fig. 7b) indicating lateral interactions between adsorbed CO\* molecules and/or adsorbed O\*, which is expected based on previous work[35]. Under non-isothermal conditions an exponential increase in rate with increasing temperature (Fig. 7d) is observed above 80 °C. It should be noted that as the surface coverage is also changing during the TPO experiment, a classic Arrhenius style analysis to calculate activation energies and pre-exponential factors is non-trivial and will be attempted in future publications. Further, we observed complex kinetic behaviour in pathway 2 below 80 °C (which was not recreated in the isothermal $O_2$ titration experiments) which cannot be described by classical Arrhenius kinetics. It could be hypothesized that some restructuring of the Pt nanoparticles is occurring around 100 °C[75], but this will be evaluated in future work combining these methods with techniques such as X-ray Absorption Spectroscopy.

The combination of TAP, kinetic modelling, and DRIFTS measurements provides unparalleled levels of kinetic insight into the dynamic site dependent activity for CO oxidation over 2 nm-sized Pt/SiO₂ catalysts. The precisely defined nature of the TAP experiment means that fine kinetic features are resolvable by modelling the exit flux curves using efficient analytical functions. By coupling this insight with DRIFTS measurements, we have been able to identify that site-specific kinetics exists for CO oxidation over 2 nm Pt nanoparticles with the three pathways for the interaction of oxygen with CO\* summarised in Fig. 8. We find two pathways for the oxidation of CO\* and one pathway for the irreversible adsorption of oxygen (no reaction). Pathway 1 mainly occurs at the UC sites and has slow kinetics, is coverage independent, and is temperature independent. On the contrary, pathway 2 mainly occurs at WC sites with fast kinetics, is highly dependent on the CO\* or O\* coverage and shows an exponential increase with temperature. These results serve as a significant insight into understanding the kinetics of various reactive sites in heterogeneous catalysis. Typically, it has been widely accepted that increasing the number of UC sites is advantageous for CO conversion[22,26]. However, even though the reaction at the UC sites is found to be barrierless, it has slow kinetics which may not be optimal for steady-state CO conversion.

The use of probe molecules to titrate catalytically active sites is commonplace in heterogeneous catalysis research[22]. Using molecules

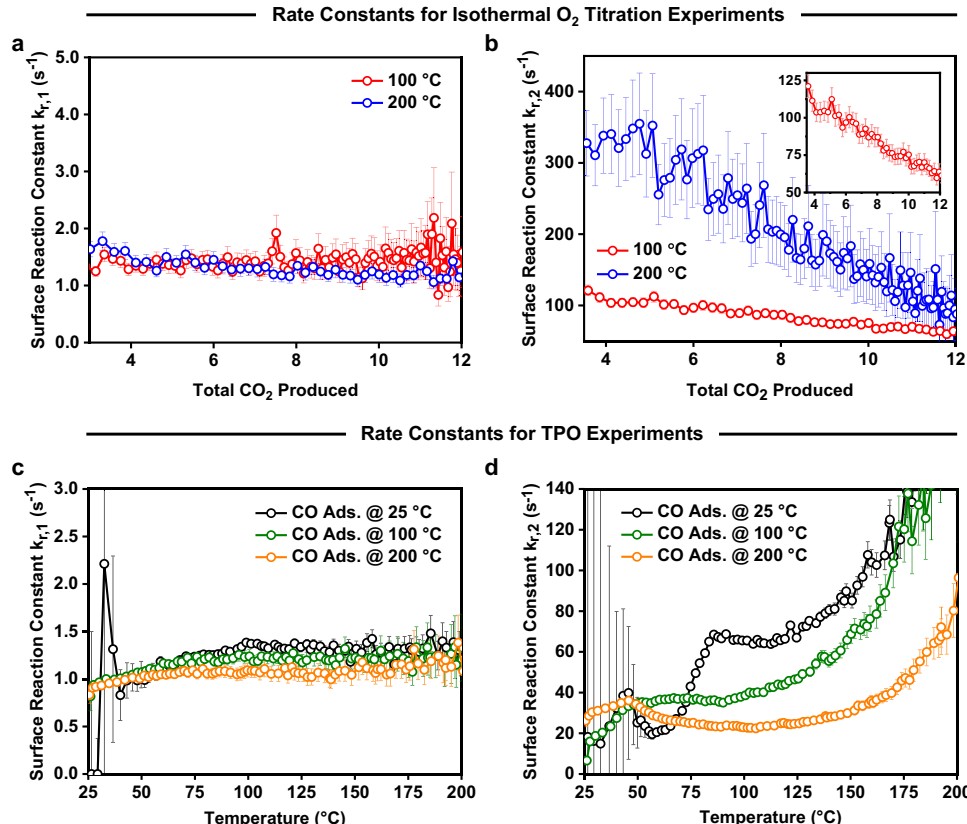

**Fig. 7 | Calculated rate constants ($k_{r,1}$) and $k_{r,2}$) for the surface reaction between adsorbed oxygen and CO from the three-pathway kinetic model with the error bars representing the 95% confidence intervals for the fitted variables.**
**a**, **b** Isothermal titration experiments where CO was adsorbed at 100 and 200 °C. Increasing total $CO_2$ produced is correlated with decreasing CO coverage. **c**, **d** TPO experiments where CO was adsorbed at temperature ranging from 25–200 °C.

When the production of $CO_2$ is sufficiently low in the TPO experiment ( > 200 °C) the signal/noise ratio of the $CO_2$ exit flux curves significantly decreases, and the oxidation state of Pt is uncertain (see Supplementary VI), which in turn decreases the confidence in the model fitting, particularly for pathway 2, as shown in Supplementary Fig. 12.

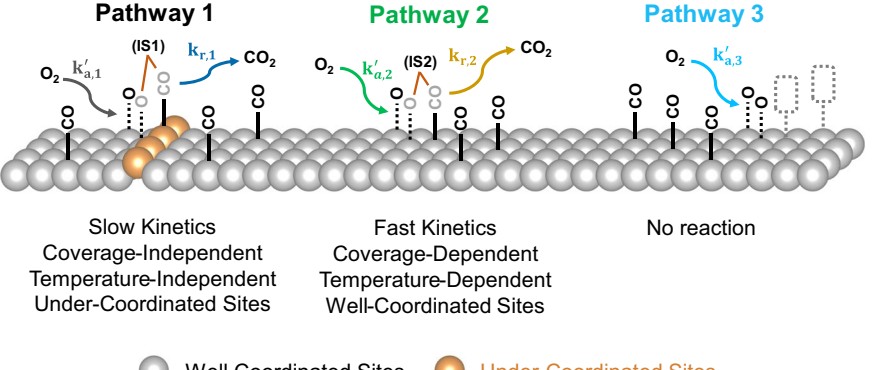

**Fig. 8 | Summary of the three-pathway model for the reaction of $O_2$ with pre-absorbed $CO^*$ over $SiO_2$-supported 2 nm Pt nanoparticles.** Pathway 1 occurs over the under-coordinated Pt sites with slow kinetics that are coverage and

temperature independent. Pathway 2 occurs over well-coordinated sites with fast kinetics that are coverage and temperature dependent. Pathway 3 which represents the irreversible adsorption/dissociation of $O_2$ occurs on the well-coordinated sites.

such as CO, $H_2$, or $N_2O$ for metal sites, $NH_3$ or pyridine for zeolites, and $O_2$ for oxygen vacancies it becomes possible to count the number of active centres on a catalyst. By performing these experiments in a TAP reactor and coupling them with Multi-Zone TAP Reactor Theory it becomes possible to not only quantify the number of sites, but simultaneously measure their catalytic activity for simple probe reactions (e.g., the oxidation of CO). As the activity for a given probe reaction is directly related to the environment in which that probe molecule is

adsorbed, the intrinsic rate constant for the rate of transformation of that species provides a fingerprint of that specific site. Using the kinetic site deconvolution method outlined in this work it then becomes possible to not only count the total number of sites, but to also count the number and distribution of different sites on the catalyst. This allows the structural characterization of catalysts allowing insight regarding size-effects (from a single atom to bulk catalyst) and support-effects, but it also opens up the possibility of rationalising how catalysts

dynamically restructure by seeing how the intrinsic rate constants of the probe reactions are modified based on the catalyst state. While this methodology is still in its infancy, we believe that this new approach is general enough to apply to other catalytic systems and can serve as a new toolkit in the characterisation of heterogeneous catalysts.

## Methods

### Preparation of Pt/SiO₂ catalyst

In a typical synthesis of 2 nm-sized Pt nanoparticles,[76] 100 mg of chloroplatinic acid hydrate ($H_2PtCl_6 \cdot xH_2O$, 99.9%, Sigma–Aldrich), 20 mg of poly(vinylpyrrolidone) (PVP, $M_w = 40000$, Sigma-Aldrich), 2.5 mL of sodium hydroxide (NaOH, 1 N) were dissolved in 10 mL of ethylene glycol (Sigma-Aldrich) in a 50 mL three-neck round-bottom flask. The flask was heated to 80 °C and evacuated for 30 min with vigorous stirring (400 rpm), then heated to 200 °C with a ramp up rate of 10 °C/min. The solution was kept at 200 °C for 2 hr with Ar gas purging. After the reaction, the solution cooled down to RT. The colloidal suspension was diluted to 50 mL of acetone and centrifuged at 6500 rpm (5.45 cm radius, 2574.3 g) for 10 min twice, repeatedly. Then, the as-synthesized Pt nanoparticles were re-dispersed in 40 mL of ethanol. Then, 10 mL of the Pt-dispersed solution was dropped onto the 0.3 g of non-nanoporous (Supplementary Fig. 13) SiO₂ powder (pretreated at 700 °C for 1 hr; 10–20 nm, Sigma-Aldrich, Product No. 637238) with vigorous stirring (400 rpm). The suspension was sonicated for 20 min, subsequently evaporating the solvent in a vacuum at 50 °C overnight. The microstructure of the Pt/SiO₂ catalyst was investigated via XRD measurement (Supplementary Fig. 1b) and shows diffraction peaks at 39.5 and 46°, corresponding to the (111) and (200) planes of reference Pt (JCPDS #04-0802). For characterisation and TAP experiments, the Pt/SiO₂ catalyst was calcined at 500 °C for 1 hr in air condition to remove majority of carbonaceous capping agent with a ramp up rate of 1 °C/min. To prepare the metallic Pt nanoparticles, the Pt/SiO₂ catalyst was reduced at 300 °C for 1 hr with a ramp up rate of 1 °C/min under 10% H₂ in Ar flow. A SiO₂ (blank) sample, was also synthesised using the exact same procedure without the addition of the Pt nanoparticles. We find that no Na contamination can be observed on both the Pt/SiO₂ and the SiO₂ (blank) samples (Supplementary Fig. 14) using X-ray photoelectron spectroscopy (XPS) measurement.

### Characterisations

The morphology and size distribution of Pt/SiO₂ catalyst were investigated by transmission electron microscopy (TEM; ARM 200 F, JEOL) at 200 kV. The catalyst loading was determined using inductively coupled plasma optical emission spectroscopy (ICP-OES; 5110 ICP-OES; Agilent). The catalyst microstructure was measured by X-ray diffractometer (XRD; D2 Phaser, Bruker) with a Cu Kα (1.5406 Å) radiation source (10 mA, 30 kV). The chemical states of catalyst was investigated by X-ray photoelectron spectroscopy (XPS; Nexsa, Thermo Fisher Scientific) using an Al Kα ($hv = 1486.7$ eV) X-ray source. All core-level spectrum was calibrated by silicon oxide (Si⁴⁺) in Si 2p to 103.5 eV. The XPS sample pretreatments were performed in the DRIFTS reactor and resultant catalyst powder was pressed onto a carbon tape mounted on a Si wafer for each XPS measurement. The N₂ adsorption isotherm at 77 K of the SiO₂ support was collected by surface characterisation analyser (3Flex, Micromeritics). The steady-state flow catalytic activity was measured in a home-built ambient pressure flow reactor that has been previously described[77] with all gas mixtures reported balanced in Ar (See Supplementary IV).

### Temporal analysis of products experiments

The TAP technique has been described extensively in the literature[40,41,78], but it is summarised here. During the TAP experiment a nanomole pulse of gas (~$10^{15}$ molecules, 108 μs pulse width) is sent into a packed bed microreactor that is held at ultra-high vacuum (<$10^{-9}$ torr). During the experiment, the pulsed gas diffuses through the packed bed via Knudsen Diffusion where it can interact with the catalyst surface. Eventually the gas diffuses out the exit of the microreactor and the exit flux of is measured via mass spectrometry. Due to the precisely defined nature of Knudsen Diffusion, the shape (and magnitude) of the exit flux curves provides highly resolved kinetic insight, in particular when coupled with kinetic modelling of the exit flux curves (see section: Modelling of Temporal Analysis of Products Pulse Responses). Further, the small pulse size means that the amount of heat imparted into the system due to the exothermic reaction is negligible.

For the TAP experiments, the Pt/SiO₂ catalyst is first calcined at 350 °C by injecting 1000 pulse sets of large O₂ pulses (160 μs pulse width) until the CO₂ signal (m/z = 44) becomes near-zero to minimise the decomposition of the remaining carbonaceous capping. After injecting a series of O₂ pulses at 350 °C, the surface of Pt catalyst was oxidised (Supplementary Fig. 15). So, before all pulse/transient response experiments, the Pt/SiO₂ catalyst is reduced at 350 °C by injecting 600 pulse sets of large H₂ pulses (160 μs pulse width) to achieve a metallic Pt surface (Supplementary Fig. 15). After the combination of the O₂ and the H₂ treatments we find that no carbonaceous species can be detected on the Pt catalyst (Supplementary Fig. 16). We utilise a home-built TAP reactor[37] where the microreactor contains a layer of commercial sand (29.7 mm; 50–70 mesh SiO₂; Sigma-Aldrich) followed by a layer of Pt/SiO₂ catalyst (5.4 mg) followed by a final layer of commercial sand (34.4 mm) for a total reactor length of 64.1 mm. The exit flux of CO, O₂, Ar, and CO₂ is monitored via mass spectrometry. The integrated exit flux of the Ar tracer is used for normalisation of all pulse experiments (see Supplementary V). As the mass spectrometer can only investigate one m/z value per pulse, multiple pulses are used to scan the whole range of m/z values and are combined to one pulse set[37]. All gas mixtures reported are balanced in Ar. Minimal activity was recorded over the SiO₂ (blank) sample (Supplementary Fig. 17) during co-pulsed experiments, with zero activity recorded during a TPO experiment, meaning the catalytic activity recorded during the TPO experiments is solely prescribed to the Pt nanoparticles. The effect of the oxidation of the Pt catalyst during the CO oxidation and TPO experiments below 200 °C can be ruled out in this work (See Supplementary VI).

### Diffused reflectance infrared fourier transform spectroscopy experiments

DRIFTS experiments were carried out in a low-temperature reaction chamber (Harrick Scientific) equipped with ZnSe windows, mounted inside the sample compartment of a Bruker Invenio FT-IR spectrometer using a Praying Mantis diffuse reflectance accessory (Harrick Scientific). The catalyst sample was prepared by pressing approximately 2 mg of 2 nm Pt/SiO₂ onto a 304 stainless-steel mesh (150 × 150 mesh). The DRIFTS reactor was loaded by placing the catalyst-containing mesh on top of approximately 110 mg of 120 grit SiC, an inert support with high thermal conductivity. There can exist large temperature gradients between the thermocouple contact in a DRIFTS reactor cell and the catalyst surface temperature exposed to the infrared beam[79]. Therefore, a thermocouple was mounted in physical contact with the bottom of the stainless-steel mesh and the temperature gradient to the catalyst surface at 350 °C was less than 20 °C as calibrated by an optical pyrometer. All DRIFTS experiments used a total volumetric flow rate of 100 sccm. Each absorbance spectrum was obtained by averaging 200 background and sample scans at a resolution of 4 cm⁻¹ using a liquid-nitrogen-cooled HgCdTe (MCT) detector, while the Praying Mantis diffuse reflectance accessory and FT-IR spectrometer was purged with dry N₂. The background measurement was acquired after the catalyst sample was annealed at 350 °C for 30 mins in 5% H₂ in Ar and cooled to 35 °C in Ar. The sample measurements were acquired after the catalyst sample was annealed at 30 °C/min in Ar to the CO adsorption temperature, the temperature was maintained for 10 mins in 0.1% CO in Ar until saturation was achieved, and the sample was cooled to 35 °C in Ar. A description of the

quantitative analysis of the DRIFTS spectra is provided in the Supplementary (see Supplementary III).

## Modelling of temporal analysis of products pulse responses

To simulate the TAP exit flux response curves, Multi-Zone TAP Reactor Theory[42,43] (MZTRT) was utilised with the catalyst zone being approximated as a Thin Zone[39] in the centre of the microreactor between two layers of inert packing. Given the large number of pulses required to saturate/titrate the catalyst surface, the experiments fall well-within the "state defining" regime as required by the MZTRT model. To perform the curve fitting first, the experimentally measured signals were normalised to the inert Ar tracer and corrected using their corresponding calibration factors (see Supplementary V). Then, the curves were further normalised to the amount of reactant gas in the pulsed mixture such that the integrated area under the reactant curve is 0 at 100% conversion and 1 at 0% conversion. Next, the diffusivity of Argon in the packed bed reactor was calculated by fitting a one-zone TAP model to the Ar exit flux curve. As the Knudsen diffusivity is proportional to $\sqrt{1/M}$ where $M$ is the molecular weight of the gas, it becomes possible to calculate the diffusivity of the reactant ($O_2$, $M = 32$) and product ($CO_2$, $M = 44$) gases by scaling the diffusivity relative to the inert gas (Ar, $M = 40$). When performing the fitting of the reactant and product curves, the diffusivities of the gases, the reactor length, and the void fraction of the reactor were all fixed, with the only variables being the rate constants for the corresponding model. It is very important to note the regression is performed on each set of exit flux response curves (i.e., exit flux plotted as a function of time) individually. Therefore, the rate constants are calculated separately at each pulse set (and catalyst state) during the experiment. All curve fitting was performed in the MATLAB environment using the lsqcurvefit function, with the 95% confidence intervals for the fitted variables evaluated using the nlparci function. A full description of the MZTRT model used in this work and how the model fitting is performed is available in Supplementary Information I, II, IV, and an example of the MATLAB script used to simulate the TAP experiments is included alongside this paper.

## Data availability

The data reported in all figures is available in the Source Data file alongside this manuscript. Additional data are available from the corresponding authors upon reasonable request. Source data are provided with this paper.

## Code availability

An example MATLAB script for the modelling of the TAP response curves is also included in the Supplementary Information alongside the data for the 25TPO experiment. The MATLAB script accompanying this paper requires the Curve Fitting Toolbox and the Statistics and Machine Learning Toolbox.

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

## Acknowledgements

C.R. gratefully acknowledges the Rowland Fellowship through the Rowland Institute at Harvard. The authors would like to acknowledge Samantha Le for their assistance generating replicate data.

## Author contributions

Taek-Seung Kim: Conceptualization, Investigation, Formal analysis, Writing – original draft Christopher R. O'Connor: Investigation, Writing – original draft Christian Reece: Conceptualization, Formal analysis, Supervision, Writing – original draft, Writing – review & editing T.S.K. performed the synthesis, characterisation and TAP experiments. C.R.O. performed the DRIFTS experiments. C.R. performed the TAP modelling and supervised the project. All authors in frequent discussions and contributed significantly to writing the manuscript.

## Competing interests

The authors declare no competing interests.
