## [Peer Review File · Nature Communications]

Interrogating Site Dependent Kinetics over SiO₂-Supported Pt NanoparticlesREVIEWER COMMENTS

Reviewer #1 (Remarks to the Author):

The manuscript of Kim et al. presents the study of the site-dependent kinetics of CO oxidation over SiO₂-supported Pt nanoparticles. The experimental results are solid, novel, and clearly described. The used methods are insightful and quantitative. I have some doubts, however, if the paper is of broad interest... maybe it can be improved by showing more concrete perspectives. Based on it, I recommend major revision. The detailed comments are listed below:

1. I am not sure that the perspectives on the use of this methodology for other catalytic systems, beyond CO oxidation, are sufficiently presented. I think it would be interesting to know what are the limitations of the methods, and for which other catalytic systems and processes they can be applied. Somehow, I do not see a direct link to operando catalysis, due to the pressure gap, but see perspectives in bridging the gap between the single crystal model catalysts and realistic powder catalysts and comparing the structure of powder catalysts (not necessarily operando) prepared by different methods and different supports.

2. Only one Pt/SiO₂ catalyst is studied. The conclusions could become more significant if the Pt particle size were varied or different supports could be compared. Currently, there is a curious result but I do not see the impact of the obtained knowledge on the catalytic science.

3. Considering the previous literature, were different pathways of CO oxidation on platinum nanoparticles ever considered or observed? What do the authors think about this paper claiming room temperature CO oxidation on Pt particles (<https://www.nature.com/articles/ncomms9675>) or maybe other literature?

4. I do not understand why the term “operando” is used in this paper. This term is typically reserved for the methods providing direct information about the structure of a catalyst. I believe, TAP with all its uniqueness still provides indirect information, as it analyses the gas phase composition, and DRIFTS is done in situ.

5. Minor comment: On page 3, it is mentioned that the reaction order in CO is 1.02. I believe it should be negative.

Reviewer #2 (Remarks to the Author):

Concept on manuscript 447948: Interrogating Site Dependent Kinetics over SiO₂-Supported Pt Nanoparticles
General concept

This is a very interesting paper that claims “Herein, we can bypass the pressure, materials, and temperature gaps, resolving and quantifying two distinct pathways for CO oxidation over SiO₂-supported 2 nm Pt nanoparticles under operando conditions.” Therefore, new venues for kinetic analysis may be open if this study is further corroborated independently. However, there are several aspects of the work that need careful consideration to properly sustain such a claim. Please find them below:

1. My main concern with the presented results is the possible influence of transport artifacts on the measurement of reaction rates and further modelling of the kinetic data. On the one hand, the authors do not provide any information whatsoever on the porosity of their catalyst. Ideally, if the authors are ascribing all catalysis to the metal, the latter should be fully exposed at the external surface of the SiO₂ support. For this, the support should either be macroporous or non-porous. Which of these instances apply here? On the other hand, CO oxidation is a highly exothermic reaction thus eliminating temperature gradients during catalytic tests is hard to accomplish. Therefore, in my opinion, the authors must present a rigorous evaluation of the possible influence of mass and heat transport limitations in the paper to validate the claims that they are making.

2. I am also concerned about the possible influence of impurities from the SiO₂ support on the catalytic activity. Particularly, commercial SiO₂ samples tend to be contaminated with traces of Na. Furthermore, the authors used NaOH for the preparation of the catalyst. The authors must then present a full chemical analysis (both bulk and surface) of the composition of the catalyst to discard undesired effects on the measured kinetics. In this sense, I couldn't find results of blank tests that serve to discard effects from the diluent of the catalyst and from the SiO₂ support. These results must be included in the Supplementary information of the paper.

3. Quantification by mass spectrometry is complex. According to the manuscript, the authors normalized the signals from the mass spectrometer with the Ar signal. However, when MS analysis is made, the amount of substance of the analyte may strongly influence quantification; particularly when the sum of the stoichiometric numbers of the reaction is not zero; which is the case for CO oxidation. This causes an increase in the pressure which combined with the fact that tests are being performed at ramped temperatures may strongly affect quantification. Therefore, the authors must discuss the latter in the experimental section of the paper and should discuss how this can affect their data analysis.

4. Although the authors made a reduction treatment of their catalyst, no evidence of the full reduction of the metal is presented. In addition, literature has presented evidence on the fact that traces of carbonaceous species may remain tightly bound to supported noble metal catalysts even after oxidation treatments. In particular, unpublished results from our lab have shown supported platinum catalysts that undergo oxidation treatments followed by reductions treatments produce methane during the latter. This can, of course, be due to impurities in the gas fed to the reactor. The authors should then ensure that such effects are not affecting their analysis.

5. According to the paper, catalytic tests were made with an O₂ to CO ratio ~ 2 which is four times higher than the stoichiometric ratio for the reaction. Why did you make this choice? What would be effect of changing this ratio on the presented conclusions? Would the two claimed mechanisms exist under both the stoichiometric and an under-stoichiometric reaction ratio?

6. The data for the apparent activation energy presented in Figure 2a suggests an alternative interpretation from the one provided by the authors. Indeed, the authors claim that all their data is fitted by a linear curve, but the dispersion of the data is strong starting around $1/RT \sim 0.282$ mol/kJ. On the

one hand, this suggests that the statistics of the data after such a value require very careful checking of influence of the correlation effects over the data. On the other hand, this indicates that reaction regime is modified after some temperature. This does not necessarily imply a new mechanism because heat transport artifacts may strongly influence the measured kinetics. And, in the case of the observed effect in Figure 2a, the existence of two different mechanisms when changing the temperature of the catalytic tests seems to contradict the claim of the authors.

Other minor concerns are:

(i) The description of the experiments is often times incomplete so one may have problems trying to reproduce the work of the authors. First, although the techniques used for the characterization of the catalyst are kind of routine for catalysis researchers, every laboratory uses some slight modifications when performing them and this hinders reproducibility. Therefore, I recommend including full details of the presented experiments in the Supplementary Information of the paper. Please, do not make the reader read other papers to understand and reproduce your experiments.

(ii) The "error" bars presented for the data in Supplementary Figures does not allow understanding the data. Please, make these plots clearer. Also, you may include a Table for reporting the statistics of the measurements.

(iii) How did you measure the flux of the products of the reactor? Typically, flux refers to a quantity per area of flow. What was the area of flow?

(iv) The first paragraph of the introduction of the paper seems to suggest that the method developed by the authors will fill the gap between laboratory catalysts and technical catalysts. I am sure that the authors are aware of the fact that the latter are multicomponent formulations whose behavior in industrial units cannot be understood only from chemical kinetics on powders. Therefore, I suggest that the authors take this into account for modifying the paragraph.

Reviewer #3 (Remarks to the Author):

Authors report the synthesis of Pt/SiO₂ catalyst using polymer stabilized polyol method. This catalyst contains ca.2nm Pt nanoparticles and these materials have been tested for CO oxidation. Fundamental studies on the kinetics of this reaction have been done in a TAP reactor. The authors have done an excellent study on the site dependency of the kinetics of CO oxidation using two different Pt nanoparticles sites (well coordinated and under coordinated sites) on a practical catalyst. The results are interesting and are of fundamental importance. It is a very well written article. Typically support materials play an important role in CO oxidation, however here the whole discussion is on Pt sites. I appreciate it is a SiO₂ support. Authors can comment on the versatility of this methodology for other catalytic systems where support play an active role in the catalysis.

Reviewer #4 (Remarks to the Author):

The manuscript reports an interesting and well-constructed kinetic and spectroscopic investigation of the

oxidation of CO on Pt supported over silica. The conclusions of the study propose that at least two parallel reaction pathways are taking place on Pt, related to sites with different coordinations. Although Pt/SiO₂ is effectively a purely academic catalyst with activity far lower than that of redox oxide-supported materials, the results underlines new methodologies to investigate this reaction and will appeal to the community.

The main shortcoming of the paper resides in the data analysis that does not take into account the potential bulk oxidation of Pt at low CO coverages or O₂ excess pulses. This could explain the third irreversible adsorption pathway for O₂ (line 214), rather than mere surface adsorption. To this respect, the work of van Bohhoven 's group on the oxidic nature of the Pt phase at low CO coverage/high temperature should be cited and discussed in the present paper (*Angew. Chem. Int. Ed.* 2008, 47, 9260–9264). The simultaneous presence of reduced and oxidised Pt phases during low temperature CO oxidation has also been reported and emphasises the complex evolution of Pt phases that may occur when reaction conditions are changed (e.g. Meunier et al, *Angew. Chem. Int. Ed.* 2021, 60, 3799 – 3805). This may thus relate to the potential changes of Pt structure as hypothesised by the authors on line 418. I therefore suggest that Pt bulk oxidation should be considered and discussed in the data analysis and model.

Minor point: line 98: CO reaction order is -1, not +1, please correct.

We would like to thank the reviewers for their feedback on the manuscript. Outside of a few clarifying questions on the oxidation state of the catalyst, we feel that the primary concern was related to the broad applicability of the kinetic site deconvolution method. In response to this we have included a new section in our manuscript discussing the applicability of this method to commonly used site-titration experiments in heterogeneous catalysis. Although this was not mentioned by the reviewers, we have also decided to rename the “bridge” CO in our DRIFTS spectra to “multi-bound” CO as the three-fold hollow site also sits within that adsorption band.

Our responses to reviewers’ specific comments are included below. The reviewers’ comments are in black, and our responses are in red. All line numbers are based on the unmarked edited manuscript.

Reviewer #1

The manuscript of Kim et al. presents the study of the site-dependent kinetics of CO oxidation over SiO₂-supported Pt nanoparticles. The experimental results are solid, novel, and clearly described. The used methods are insightful and quantitative. I have some doubts, however, if the paper is of broad interest... maybe it can be improved by showing more concrete perspectives. Based on it, I recommend major revision. The detailed comments are listed below:

We thank reviewer #1 for their interest in our paper and constructive comments. As this is a new method of quantifying active sites, we can understand their concerns with respect to applicability. We have included a new section of the paper (see below) where we further discuss the future perspectives of this method and also provide candidates for future applications.

I am not sure that the perspectives on the use of this methodology for other catalytic systems, beyond CO oxidation, are sufficiently presented. I think it would be interesting to know what are the limitations of the methods, and for which other catalytic systems and processes they can be applied. Somehow, I do not see a direct link to operando catalysis, due to the pressure gap, but see perspectives in bridging the gap between the single crystal model catalysts and realistic powder catalysts and comparing the structure of powder catalysts (not necessarily operando) prepared by different methods and different supports.

We agree with reviewer #1 that the term *operando* was misused. With regards to the pressure gap, the peak pressure during a pulse in the TAP reactor reaches approximately 1 mbar, meaning it is near ambient pressure. Further, pre-treatments of the catalyst are performed using large pulsing where the peak pressure in the microreactor reaches ≥ 1 bar. We agree that the experiment is not within the specific subclass of *operando* measurements, but we feel that it can be considered analogous to *in situ* methods and have reworded the paper as such.

In response to the broad applicability of the method, we have included a new section in the paper outlining future perspectives on the use of this methodology. We would like to emphasise that use of probe molecules for characterising heterogeneous catalysts has been commonplace for many years. We acknowledge that CO oxidation is not a catalytically relevant process, but it is an excellent tool for probing catalytic activity and for determining the number of active sites. Probe molecules such as CO for metal sites, ammonia/pyridine for acid sites in zeolites, or oxidants (e.g., O₂ NO₂) to probe reducible oxides/catalytic centres in zeolites can be readily utilized in this method. The reactions of these species occur via (relatively) simple first-order relationships meaning that the active-site deconvolution method we have developed is directly applicable. In fact, we aim to broadly apply this method to new materials in the future. We acknowledge that this methodology is still in the early stages of application, but it is our hope that many others will utilise it in the further for more complex systems.

Lines 456 – 472 now read:

“The use of probe molecules to titrate catalytically active sites is commonplace in heterogeneous catalysis research¹. Using molecules such as CO, H₂, or N₂O for metal sites, NH₃ or

pyridine for zeolites, and O₂ for oxygen vacancies it becomes possible to count the number of active centres on a catalyst. By performing these experiments in a TAP reactor and coupling them with Multi-Zone TAP Reactor Theory it becomes possible to not only quantify the number of sites, but simultaneously measure their catalytic activity for simple probe reactions (e.g., the oxidation of CO). As the activity for a given probe reaction is directly related to the environment in which that probe molecule is adsorbed, the intrinsic rate constant for the rate of transformation of that species provides a fingerprint of that specific site. Using the kinetic site deconvolution method outlined in this work it then becomes possible to not only count the total number of sites, but to also count the number and distribution of different sites on the catalyst. This allows the structural characterization of catalysts allowing insight regarding size-effects (from a single atom to bulk catalyst) and support-effects, but it also opens up the possibility of rationalising how catalysts dynamically restructure by seeing how the number of sites and the intrinsic rate constants of the probe reactions are modified based on the catalyst state. While this methodology is still in its infancy, we believe that this new approach is general enough to apply to other catalytic systems and can serve as a new toolkit in the characterisation of heterogeneous catalysts.”

Only one Pt/SiO₂ catalyst is studied. The conclusions could become more significant if the Pt particle size were varied or different supports could be compared. Currently, there is a curious result but I do not see the impact of the obtained knowledge on the catalytic science.

Ideally, it would be great to study different size Pt nanoparticles, but I’m sure the reviewer can sympathise that generating well-defined Pt nanoparticles with a narrow size distribution at different sizes *that also don’t sinter/restructure under reaction conditions* is very difficult. The narrow size distribution is mandatory otherwise it is not feasible to prescribe catalytic activity to specific sites. As developing the kinetic deconvolution method is already a significant body of work that could be directly applicable to other catalytic systems, we feel that even with the one case study the work warrants publication. However, we would like to mention that future work studying the effect of Pt nanoparticle size is currently underway.

Considering the previous literature, were different pathways of CO oxidation on platinum nanoparticles ever considered or observed? What do the authors think about this paper claiming room temperature CO oxidation on Pt particles (<https://www.nature.com/articles/ncomms9675>) or maybe other literature?

The paper mentioned claims that room temperature CO oxidation on Pt/Al₂O₃ occurs on isolated “oxidic” Pt centres through formation of a Pt carbonate intermediate. While the paper is interesting, we do not feel that it is relevant to our work as the formation of these centres seem specific to Al₂O₃ supported Pt catalysts whereas we used SiO₂ as a support material. Further, we see no evidence of Pt single atoms or the formation of Pt(CO₃) species in the DRIFTS spectra on our catalyst, but we also cannot completely rule out this mechanism.

We would also like to emphasise that *our methodology doesn’t suggest a specific pathway*, and it is just simply a measurement of an intrinsic rate constant for a given elementary reaction step. While those intrinsic rate constants can provide insight into the nature of that active site (the rate, coverage dependence, temperature dependence) we cannot suggest a specific reaction mechanism without further characterisation methods / experiments / theory. Our rationale for the site-specificity is due to the strong correlation between the estimated distribution of under-coordinated and well-coordinated sites and the distribution of distinct CO oxidation sites on the surface, but how that reaction proceeds on that site is somewhat ambiguous. We recognise that this required further clarification and so have expanded upon our discussion of the oxidation of CO over the undercoordinated sites.

Lines 388 – 390 now read:

“While it is now possible to identify where the reaction is occurring, the intrinsic rate constants alone cannot provide specifics on how the reaction is occurring without further characterisation methods.”

Lines 400 – 403 now read:

“Other experiments have reported that the barrierless (or near barrierless) oxidation of CO can occur via a Pt(CO₃) intermediates², or through an Eley-Rideal type mechanism³. While we do not find any direct evidence of those pathways, it is also not possible to entirely rule them out.”

I do not understand why the term “operando” is used in this paper. This term is typically reserved for the methods providing direct information about the structure of a catalyst. I believe, TAP with all its uniqueness still provides indirect information, as it analyses the gas phase composition, and DRIFTS is done in situ.

We agree with reviewer #1 that this is a valid comment. We have removed the term *operando* / changed it to *in situ* where applicable. We believe that the presented *kinetic site deconvolution* as an *in situ* characterisation method, but we recognise that is contentious and so have refrained from it’s use in this paper.

Minor comment: On page 3, it is mentioned that the reaction order in CO is 1.02. I believe it should be negative.

We thank reviewer #1 for pointing out this mistake. We have corrected the reaction order in CO to -1.02.

Reviewer #2

This is a very interesting paper that claims “Herein, we can bypass the pressure, materials, and temperature gaps, resolving and quantifying two distinct pathways for CO oxidation over SiO₂-supported 2 nm Pt nanoparticles under operando conditions.” Therefore, new venues for kinetic analysis may be open if this study is further corroborated independently. However, there are several aspects of the work that need careful consideration to properly sustain such a claim. Please find them below:

We thank reviewer #2 for their positive feedback on the manuscript.

My main concern with the presented results is the possible influence of transport artifacts on the measurement of reaction rates and further modelling of the kinetic data. On the one hand, the authors do not provide any information whatsoever on the porosity of their catalyst. Ideally, if the authors are ascribing all catalysis to the metal, the latter should be fully exposed at the external surface of the SiO₂ support. For this, the support should either be macroporous or non-porous. Which of these instances apply here?

We understand reviewer #2’s concerns. We have included BET results that confirm that our support material is nonporous/macroporous and included it in Supplementary Fig. 13.

Lines 484 – 486 now read:

“the 0.3 g of nonporous (Supplementary Fig. 13) SiO₂ powder (pretreated at 700 °C for 1 hr; 10–20 nm, Sigma-Aldrich, Product No. 637238) with vigorous stirring (400 rpm).

Supplementary Figure 13. N₂ adsorption isotherm plot for SiO₂ support material demonstrating nonporous/macroporous behaviour.

On the other hand, CO oxidation is a highly exothermic reaction thus eliminating temperature gradients during catalytic tests is hard to accomplish. Therefore, in my opinion, the authors must present a rigorous evaluation of the possible influence of mass and heat transport limitations in the paper to validate the claims that they are making.

We understand why reviewer #2 may have concerns regarding temperature gradients and mass/heat transport limitations as these are very important in steady-state flow experiments. However, one benefit of the TAP experiment is that these factors are significantly minimised.

In the TAP experiment the primary transport mechanism is Knudsen diffusion which is well-defined and independent of composition of the gas mixture and its pressure. This has been discussed extensively in previous publications, most notably in the seminal work of Gleaves ([https://doi.org/10.1016/S0926-860X\(97\)00124-5](https://doi.org/10.1016/S0926-860X(97)00124-5))⁴ but also by Constaes in their derivations of Multi-Zone TAP Reactor Theory ([https://doi.org/10.1016/S0009-2509\(00\)00216-5](https://doi.org/10.1016/S0009-2509(00)00216-5))⁵. Further, even though CO oxidation is highly exothermic, the total amount of reactive gas in a single pulse is very small (~1nmol) which means even in the extreme case of every CO molecule instantaneously reacting, the total amount of heat imparted into the system is multiple orders of magnitude than the specific heat capacity of our catalyst. We have added an extra statement clarifying that we are not heat/mass transfer limited under these conditions.

Lines 520 – 522 now read:

“As the primary transport mechanism is Knudsen diffusion any mass transfer effects can be minimised. Further, the small pulse size means that the amount of heat imparted into the system due to the exothermic reaction is negligible.”

I am also concerned about the possible influence of impurities from the SiO₂ support on the catalytic activity. Particularly, commercial SiO₂ samples tend to be contaminated with traces of Na. Furthermore, the authors used NaOH for the preparation of the catalyst. The authors must then present a full chemical analysis (both bulk and surface) of the composition of the catalyst to discard undesired effects on the measured kinetics. In this sense, I couldn't find results of blank tests that serve to discard effects from the diluent of the catalyst and from the SiO₂ support. These results must be included in the Supplementary information of the paper.

Reviewer #2 raises a good point. While we have performed extensive blank tests over the SiO₂ sand used as the inert packing in the TAP reactor, we did not report them as part of the paper. To ensure that no catalytic activity was coming from the SiO₂ support material (and the inert SiO₂ sand), we have synthesised a blank sample using same synthetic procedure (including the addition of NaOH) as that of the Pt/SiO₂ catalyst but without including the Pt nanoparticles which we call SiO₂ (blank). We investigated the chemical states of the SiO₂ (blank) sample and Pt/SiO₂ catalyst using XPS. We consider the XPS measurement to be a bulk chemical analysis for the 2 nm Pt nanoparticles because the XPS probing depth is ~3 nm using an Al K α X-ray source. For the both samples, there is no peak for Na compounds in the Na 1s core-level spectrum (Supplementary Fig. 14).

Supplementary Figure 14. a,b, XPS measurements for Na 1s core-level spectrum of (a) SiO₂ (blank) sample and Pt/SiO₂ catalyst.

However, to further confirm that the catalytic activity is primarily from the Pt nanoparticles, the SiO₂ (blank) sample is packed in the exact same configuration as used during the experiments and the CO oxidation and TPO pulsed experiments were repeated (Supplementary Fig. 17). Little-to-no activity for CO oxidation is recorded on the blank sample.

Supplementary Figure 17. a, Temperature-dependent integrated exit flux of $m/z = 44$ (CO_2) normalised *via* Ar on CO^* -covered Pt/SiO_2 catalyst and SiO_2 (blank) for CO oxidation (6.6% CO 13.4% O_2 gas mixture in an inert Ar tracer) while heating from RT–350 °C at a heating rate of 8 °C/min. **b**, Integrated Ar normalised exit flux of $m/z = 44$ (CO_2) during TPO experiments on the CO^* -covered Pt/SiO_2 catalyst and SiO_2 (blank) where CO^* was preadsorbed at RT. Then O_2 was repeatedly pulsed over the catalyst while being linearly heated to 350 °C at 8 °C/min.

Lines 494 – 497 now read:

“A SiO_2 (blank) sample, was also synthesised using the exact same procedure without the addition of the Pt nanoparticles. We find that no Na contamination can be observed on both the Pt/SiO_2 and the SiO_2 (blank) samples (Supplementary Fig. 14) using XPS.”

Lines 539 – 541 now read:

“Minimal activity was recorded over the SiO_2 (blank) sample (Supplementary Fig. 17) meaning the catalytic activity recorded during the TAP experiments is solely prescribed to the Pt nanoparticles.”

Quantification by mass spectrometry is complex. According to the manuscript, the authors normalized the signals from the mass spectrometer with the Ar signal. However, when MS analysis is made, the amount of substance of the analyte may strongly influence quantification; particularly when the sum of the stoichiometric numbers of the reaction is not zero; which is the case for CO oxidation. This causes an increase in the pressure which combined with the fact that tests are being performed at ramped temperatures may strongly affect quantification. Therefore, the authors must discuss the latter in the experimental section of the paper and should discuss how this can affect their data analysis.

It is important to note that in TAP experiment we are not measuring concentration of a gas stream, but instead we are measuring the flux of gas at the reactor exit during a vacuum pressure pulse experiment. Therefore, the non-zero stoichiometry for the reaction is not relevant and does not affect the data analysis. Further, the quantity of reactive gas in each pulse is known by normalising the signal to the inert Ar tracer (which remains at a fixed amount). By comparing the Ar normalised integrated exit flux for the reactive species (O_2 , CO, CO_2) to standards measured by pulsing known mixtures of gas over a bed packed with inert sand, it becomes possible to quantify how much reactant/product gas is produced in each pulse. This is outlined in section V of the supplementary information. We realised that the term concentration is misused in the section: Modelling of Temporal Analysis of Products Pulse Responses and have changed **line 571** to: “Then, the curves were further normalised to the amount of reactant gas...”.

In the flow reactor experiments, the total concentration of reactants in the gas stream is 7.5% (balanced in Ar) the maximum conversion is 13% giving <1% maximum error in the total relative concentration due to the non-stoichiometric reaction. This is well within the noise of the mass spectrometer signal and so does not affect our results.

We have included the following statement in the caption of Supplementary Fig. 2.

“The total conversion of reactants relative to the entire gas stream is sufficiently low that any change in relative concentration due to the non-stoichiometric reaction is within the noise of the mass spectrometer signal.”

Although the authors made a reduction treatment of their catalyst, no evidence of the full reduction of the metal is presented.

We understand reviewer #2's concern regarding the oxidation state of the Pt sample. The combination of an oxidative and reductive treatment at elevated temperatures should be sufficient enough to ensure the Pt is metallic based on previous surface science literature^{6,7}. Further, our CO-DRIFTS spectra match that of metallic Pt, and we record no PtO_x-CO binding modes. However, we recognise that perhaps more direct evidence is required. We would also like to emphasise that the H₂ treatment is repeated before each experiment to ensure that catalyst is returned to its original state.

We have performed supplemental XPS to identify the oxidation state of the 2 nm Pt/SiO₂ catalyst after O₂ treatment and after a subsequent H₂ treatment at 350 °C in Supplementary Fig. 15. The H₂-treated 2 nm Pt/SiO₂ catalyst shows the dominant metallic state of Pt (Pt⁰), according to the previous literature regarding oxidation states of ultra-small sized metallic Pt nanoparticles *via* XPS analysis⁸. Lastly, we have investigated additional TAP experiments over the oxidised Pt catalyst which is prepared *via* O₂ treatment at 350 °C. It shows a higher portion of cationic Pt²⁺ species than that of the H₂-pretreated catalyst as shown in Supplementary Fig. 15. Interestingly, the oxidised Pt catalyst shows completely different catalytic behaviour in Supplementary Fig. 18, indicative of higher catalytic activity at low temperatures (*i.e.*, a lower onset temperature of CO* conversion) and less amount of total CO* intake. Thus, we conclude that the H₂-treated Pt/SiO₂ catalyst has a metallic Pt surface.

Supplementary Figure 15. XPS measurements for Pt 4f core-level spectrum of the Pt/SiO₂ catalyst after O₂ treatment and after a subsequent H₂ treatment at 350 °C.

Supplementary Figure 18. a, Flow chart about the description of four catalysts; the Pt/SiO₂ catalysts prepared by (1) H₂, (2) O₂ pretreatment, CO oxidation (RT–350°C–RT), and CO oxidation (RT–200 °C–RT) are regards to metallic Pt, bulk oxidized Pt, spent Pt, and partially spent Pt catalysts, respectively. **b**, Temperature-dependent integrated exit flux of *m/z* = 44 (CO₂) normalised *via* Ar on the CO*–covered Pt/SiO₂ catalysts (the metallic Pt, bulk oxidised Pt, spent Pt, and partially spent Pt) for CO oxidation (6.6% CO 13.4% O₂ gas mixture in an inert Ar tracer); CO* was preadsorbed at 25 °C while heating from RT–200 °C at a heating rate of 8 °C/min.

Lines 526 – 530 now read:

“After injecting a series of O₂ pulses at 350 °C, the surface of Pt catalyst was oxidised (Supplementary Fig. 15). So, before all pulse/transient response experiments, the Pt/SiO₂ catalyst is reduced at 350 °C by injecting 600 pulse sets of large H₂ pulses (160 μs pulse width) to achieve a metallic Pt surface (Supplementary Fig. 15).”

Lines 541 – 542 now read:

“The effect of the oxidation of the Pt catalyst during the CO oxidation and TPO experiments below 200 °C can be ruled out in this work (See Supplementary VI).”

In addition, literature has presented evidence on the fact that traces of carbonaceous species may remain tightly bound to supported noble metal catalysts even after oxidation treatments. In particular, unpublished results from our lab have shown supported platinum catalysts that undergo oxidation

treatments followed by reductions treatments produce methane during the latter. This can, of course, be due to impurities in the gas fed to the reactor. The authors should then ensure that such effects are not affecting their analysis.

We agree that it is challenging to completely remove carbonaceous species from the Pt catalyst, particularly when only using an oxidative treatment. As predicted, we see a trace amount of CH₄ produced when pulsing H₂ over the O₂ treated catalyst (Supplementary Fig. 16a), but after our H₂ pretreatment we see that the CH₄ signal is completely removed (Supplementary Fig. 16b). As we always perform a H₂ treatment after the oxidative treatments, this is a very strong indication that any carbonaceous species on the catalyst are not affecting our analysis.

Supplementary Figure 16. a,b, Argon normalised exit flux curves of $m/z = 16$ (CH₄) for a pulse set of 20% of H₂ gas in an inert Ar tracer at 350 °C over oxidised Pt/SiO₂ catalyst (underwent O₂-treatment at 350 °C) (a) before and (b) after H₂ treatment. The lack of CH₄ production indicates that no carbonaceous species can be detected on the catalyst after the combined O₂ and H₂ treatments.

Lines 530 – 531 now read:

“After the combination of the O₂ and the H₂ treatments we find that no carbonaceous species can be detected on the Pt catalyst (Supplementary Fig. 16).”

According to the paper, catalytic tests were made with an O₂ to CO ratio ~ 2 which is four times higher than the stoichiometric ratio for the reaction. Why did you make this choice? What would be effect of changing this ratio on the presented conclusions? Would the two claimed mechanisms exist under both the stoichiometric and an under-stoichiometric reaction ratio?

We chose an O₂ to CO ratio ~2 to investigate the catalytic behaviour of pre-adsorbed CO* during reaction. The excess O₂ in the reactant can be used to act as a titrant for pre-adsorbed CO* and minimises CO re-adsorption to effectively probe the catalytic reactivity as a function of CO* coverage. In short, this is a TPO experiment while under reaction conditions. Additional isotope experiments were performed using a stoichiometric 2:1 CO:O₂ ratio and CO rich 3:1 CO:O₂ ratio (Fig. R1), but under these conditions the adsorption of ¹²CO inhibits the conversion of preadsorbed ¹³CO* by maintaining a high coverage during the reaction. Further, we find that the ¹²CO can place exchange with the adsorbed ¹³CO* (Fig. R2). As we were interested in probing the reactivity as a function of CO* coverage, the CO:O₂ ratio of 1:2 was chosen. Further analysis on this dataset is being performed as it is very interesting, but it is outside the scope of this work which is focused on identifying the active sites for the reaction between CO* and O*.

Figure R1. a,b, Temperature-dependent integrated exit flux of $m/z = 44$ (CO_2) and $m/z = 45$ ($^{13}\text{CO}_2$) normalised *via* Ar from the TAP experiment where (a) stoichiometric (2:1 molar ratio, 13.3% CO 6.6% O₂), and (b) carbon monoxide rich (CO/O₂ = 3:1 molar ratio, 15.0% CO 5.0% O₂) were pulsed over a $^{13}\text{CO}^*$ -covered Pt/SiO₂ catalyst while heating from RT–350 °C at a heating rate of 8 °C/min.

Figure R2. a, Integrated Ar normalised exit flux of $m/z = 29$ (^{13}CO) during the ^{13}CO adsorption *via* consecutive ^{13}CO pulses at 100 °C. **b**, Integrated Ar normalised exit flux of $m/z = 29$ (^{13}CO) and $m/z = 28$ (^{12}CO) during the consecutive ^{12}CO pulsing over the $^{13}\text{CO}^*$ -covered Pt/SiO₂ catalyst at 100 °C.

The data for the apparent activation energy presented in Figure 2a suggests an alternative interpretation from the one provided by the authors. Indeed, the authors claim that all their data is fitted by a linear curve, but the dispersion of the data is strong starting around $1/RT \sim 0.282$ mol/kJ. On the one hand, this suggests that the statistics of the data after such a value require very careful checking of influence of the correlation effects over the data. On the other hand, this indicates that reaction regime is modified after some temperature. This does not necessarily imply a new mechanism because heat transport artifacts may strongly influence the measured kinetics. And, in the case of the observed effect in Figure 2a, the existence of two different mechanisms when changing the temperature of the catalytic tests seems to contradict the claim of the authors.

The spread in the data at 0.298 mol/KJ (130 °C) to 0.282 mol/KJ (153 °C) is an artifact of the noise present in the mass spec signal at low CO conversions (Supplementary Fig. 2a). The appearance of any poor fit to the linear curve over this temperature range cannot be confidently attributed as a change reaction mechanism as suggested. We have included the temperature programmed reaction spectra in

Supplementary Fig. 2a to demonstrate the large noise in the QMS data compared to the low CO conversion for the corresponding temperature range as the apparent activation energy data.

Supplementary Figure 2. **a**, Temperature programmed reaction spectra for CO oxidation over 2 nm Pt/SiO₂ from the rate of CO consumption using a heating rate of 5 °C/min (100 ml/min total flow; 2.5% CO, 5% O₂) from 100 to 170 °C. **b**, Determination of the apparent activation energy from 130 to 160 °C of data in (a). **c,d**, Determination of the reaction order in (c) CO (1.5–3.5% CO, 5 % O₂) and (d) O₂ (2.5% CO, 3–7% O₂) from the rate of CO₂ production at 160 °C. The rate of CO₂ production was used for determining the reaction orders because of the higher sensitivity to CO₂ production than CO consumption at low conversions. The total conversion of reactants relative to the entire gas stream is sufficiently low that any change in relative concentration due to the non-stoichiometric reaction is within the noise of the mass spectrometer signal.

Other minor concerns are:

(i) The description of the experiments is often times incomplete so one may have problems trying to reproduce the work of the authors. First, although the techniques used for the characterization of the catalyst are kind of rutinary for catalysis researchers, every laboratory uses some slight modifications when performing them and this hinders reproducibility. Therefore, I recommend including full details of the presented experiments in the Supplementary Information of the paper. Please, do not make the reader read other papers to understand and reproduce your experiments.

We have expanded upon our description of the experiments and provided more specificities as requested. We have also included a new section in the supplemental (IV) which outlines the specific way we pack our flow reactor system to assist with reproducibility. While we recognise it is frustrating to have to read papers to understand experimental methods, we hope the reviewer can sympathise that this paper

is already significant in length and as such a full description of every component used in our systems is far outside the scope of this work.

(ii) The “error” bars presented for the data in Supplementary Figures does not allow understanding the data. Please, make these plots clearer. Also, you may include a Table for reporting the statistics of the measurements.

We would like to specify that the bars represent the 95% confidence intervals from the regression of the rate constants as mentioned in “Modelling of Temporal Analysis of Products Experiments” **lines 582 – 584**. We do recognise that they are difficult to see due to the transparency (particularly after being compressed) and so we have made them less transparent and clearer.

(iii) How did you measure the flux of the products of the reactor? Typically, flux refers to a quantity per area of flow. What was the area of flow?

The flux of the gas at the reactor exit is measured by a mass spectrometer mounted directly beneath the microreactor. The area of flow is the surface area of the exit of the microreactor.

(iv) The first paragraph of the introduction of the paper seems to suggest that the method developed by the authors will fill the gap between laboratory catalysts and technical catalysts. I am sure that the authors are aware of the fact that the latter are multicomponent formulations whose behavior in industrial units cannot be understood only from chemical kinetics on powders. Therefore, I suggest that the authors take this into account for modifying the paragraph.

Industrial catalysts are indeed far more complicated than laboratory catalysts. However, if the sample can be pelletised and crushed into a powder, then there is no practical reason why it cannot be placed into the TAP reactor. While the analysis would be complex, it is not impossible.

While catalytic behaviour cannot be purely understood from chemical kinetics on powders, it is the chemical kinetics itself that is the foundation of any complex modelling of an industrial reactor as it is the chemistry that largely drives the heat and mass transfer limitations. However, we recognise that the entire process is more complex than simply kinetics and on **line 26** have reworded industrial processes to industrial reactions.

Reviewer #3

Authors report the synthesis of Pt/SiO₂ catalyst using polymer stabilized polyol method. This catalyst contains ca.2nm Pt nanoparticles and these materials have been tested for CO oxidation. Fundamental studies on the kinetics of this reaction have been done in a TAP reactor. The authors have done an excellent study on the site dependency of the kinetics of CO oxidation using two different Pt nanoparticles sites (well coordinated and under coordinated sites) on a practical catalyst. The results are interesting and are of fundamental importance. It is a very well written article. Typically support materials play an important role in CO oxidation, however here the whole discussion is on Pt sites. I appreciate it is a SiO₂ support. Authors can comment on the versatility of this methodology for other catalytic systems where support play an active role in the catalysis.

We thank the reviewer for their positive feedback on the manuscript. We agree that a broader perspective on the application of the method is required, and we have expanded the manuscript to include one.

Lines 456 – 472 now read:

“The use of probe molecules to titrate catalytically active sites is commonplace in heterogeneous catalysis research¹. Using molecules such as CO, H₂, or N₂O for metal sites, NH₃ or pyridine for zeolites, and O₂ for oxygen vacancies it becomes possible to count the number of active centres on a catalyst. By performing these experiments in a TAP reactor and coupling them with Multi-Zone TAP Reactor Theory it becomes possible to not only quantify the number of sites, but simultaneously measure their catalytic activity for simple probe reactions (e.g., the oxidation of CO). As the activity for a given probe reaction is directly related to the environment in which that probe molecule is adsorbed, the intrinsic rate constant for the rate of transformation of that species provides a fingerprint of that specific site. Using the kinetic site deconvolution method outlined in this work it then becomes possible to not only count the total number of sites, but to also count the number and distribution of different sites on the catalyst. This allows the structural characterization of catalysts allowing insight regarding size-effects (from a single atom to bulk catalyst) and support-effects, but it also opens up the possibility of rationalising how catalysts dynamically restructure by seeing how the number of sites and the intrinsic rate constants of the probe reactions are modified based on the catalyst state. While this methodology is still in its infancy, we believe that this new approach is general enough to apply to other catalytic systems and can serve as a new toolkit in the characterisation of heterogeneous catalysts.”

Reviewer #4

The manuscript reports an interesting and well-constructed kinetic and spectroscopic investigation of the oxidation of CO on Pt supported over silica. The conclusions of the study propose that at least two parallel reaction pathways are taking place on Pt, related to sites with different coordinations. Although Pt/SiO₂ is effectively a purely academic catalyst with activity far lower than that of redox oxide-supported materials, the results underlines new methodologies to investigate this reaction and will appeal to the community.

We thank the reviewer for their positive feedback on the manuscript. Pt/SiO₂ is certainly an academic catalyst, and (perhaps ironically) we chose the catalyst to develop our kinetic site deconvolution technique due to it's supposed "simplicity"!

The main shortcoming of the paper resides in the data analysis that does not take into account the potential bulk oxidation of Pt at low CO coverages or O₂ excess pulses. This could explain the third irreversible adsorption pathway for O₂ (line 214), rather than mere surface adsorption. To this respect, the work of van Bokhoven's group on the oxidic nature of the Pt phase at low CO coverage/high temperature should be cited and discussed in the present paper (*Angew. Chem. Int. Ed.* 2008, 47, 9260–9264). The simultaneous presence of reduced and oxidised Pt phases during low temperature CO oxidation has also been reported and emphasises the complex evolution of Pt phases that may occur when reaction conditions are changed (e.g. Meunier et al, *Angew. Chem. Int. Ed.* 2021, 60, 3799 – 3805). This may thus relate to the potential changes of Pt structure as hypothesised by the authors on line 418.

I therefore suggest that Pt bulk oxidation should be considered and discussed in the data analysis and model.

We agree with reviewer #4 that the bulk oxidation of Pt is of extreme importance and thank them for their very insightful feedback. As described in both papers (*Angew. Chem. Int. Ed.* 2008, 47, 9260–9264 and *Angew. Chem. Int. Ed.* 2021, 60, 3799–3805), the reactivity for CO oxidation over the Pt/SiO₂ is quite different on the oxidised and metallic Pt surface. The O₂-rich reaction gas mixture (O₂/CO = 2) in this work has a similar ratio to the condition that the reviewer mentioned (low CO coverages and excess O₂ molecules), which potentially makes the bulk oxidation of Pt relevant under steady-state conditions.

We have performed additional TAP experiments over the oxidised Pt catalyst which is prepared *via* O₂ treatment at 350 °C. It shows a higher portion of cationic Pt²⁺ species than that of the H₂-treated catalyst (see Supplementary Fig. 15). As we expected, the oxidised Pt catalyst shows completely different catalytic behaviour compared to metallic Pt (see Supplementary Fig. 18), demonstrating higher catalytic activity at low temperatures (*i.e.*, a lower onset temperature of CO* conversion). Notably, based on their distinguishable catalytic activity on the oxidised and metallic Pt surface, we can use this to approximate the oxidation state of Pt (*e.g.*, bulk oxidised, partially oxidized, and metallic). To determine when oxidation begins during the CO reaction, we measured the catalytic performance for CO oxidation over the CO*-covered Pt/SiO₂ for the spent Pt (25–350 °C) and partially spent Pt (25–200 °C). The spent Pt shows similar results to that of the bulk oxidised Pt and the partially spent Pt exhibits similar results to that of the metallic Pt. Therefore, we conclude that Pt nanoparticles are metallic until at least 200 °C. Although this is not a direct spectroscopic measurement, it can serve as an indicator to define surface states via their distinguishable reactivities. Thus, we can justify our methodology is effective up to at least 200 °C without oxidation effect. In the original paper draft in the caption of **Figure 7** we mention that “When the production of CO₂ is sufficiently low in the TPO experiment (> 200 °C) the signal/noise ratio of the CO₂ exit flux curves significantly decreases, which in turn decreases the confidence in the model fitting, particularly for pathway 2, as shown in Supplementary Fig. 12.” in the caption of Fig. 7. We now think the oxidation of Pt most likely also plays a role in the decreased confidence of the model fitting and have adjusted the caption for **Figure 7**, so it now reads:

“When the production of CO₂ is sufficiently low in the TPO experiment (> 200 °C) the signal/noise ratio of the CO₂ exit flux curves significantly decreases, and the oxidation state of Pt is uncertain (see Supplementary VI), which in turn decreases the confidence in the model fitting, particularly for pathway 2, as shown in Supplementary Fig. 12.”

The Pt catalyst almost certainly gets oxidised at high temperature (> 200 °C) in the TPO experiment, but as H₂ treatments were performed between every experiment we rule out the oxidation effect in this work.

Regarding line 418, as we see now have evidence of no direct oxidation of the Pt below 200 °C we do not believe that this is facilitating the change in rate constant that we observe at 100 °C.

Supplementary Figure 15. XPS measurements for Pt 4f core-level spectrum of the Pt/SiO₂ catalyst treated in H₂ and O₂ at 350 °C.

Lines 540 – 541 now read:

“The effect of the oxidation of the Pt catalyst during the CO oxidation and TPO experiments below 200 °C can be ruled out in this work (See Supplementary VI).”

Supplementary Figure 18. **a**, Flow chart about the description of four catalysts; the Pt/SiO₂ catalysts prepared by (1) H₂, (2) O₂ pretreatment, CO oxidation (RT–350°C–RT), and CO oxidation (RT–200 °C–RT) are regards to metallic Pt, bulk oxidized Pt, spent Pt, and partially spent Pt catalysts, respectively. **b**, Temperature-dependent integrated exit flux of m/z = 44 (CO₂) normalised *via* Ar on the CO*–covered Pt/SiO₂ catalysts (the metallic Pt, bulk oxidised Pt, spent Pt, and partially spent Pt) for CO oxidation (6.6% CO 13.4% O₂ gas mixture in an inert Ar tracer); CO* was preadsorbed at 25 °C while heating from RT–200 °C at a heating rate of 8 °C/min.

Minor point: line 98: CO reaction order is -1, not +1, please correct.

We thank reviewer #4 for pointing out this mistake. We have corrected the reaction order in CO to -1.02.

1. Vogt, C. & Weckhuysen, B. M. The concept of active site in heterogeneous catalysis. *Nat. Rev. Chem.* **6**, 89–111 (2022).
2. Newton, M. A., Ferri, D., Smolentsev, G., Marchionni, V. & Nachtegaal, M. Room-temperature carbon monoxide oxidation by oxygen over Pt/Al₂O₃ mediated by reactive platinum carbonates. *Nat. Commun.* **6**, 8675 (2015).
3. Allian, A. D. *et al.* Chemisorption of CO and mechanism of CO oxidation on supported platinum nanoclusters. *J. Am. Chem. Soc.* **133**, 4498–4517 (2011).
4. Gleaves, J. T., Yablonskii, G. S., Phanawadee, P. & Schuurman, Y. TAP-2: An interrogative kinetics approach. *Appl. Catal. A-Gen.* **160**, 55–88 (1997).
5. Constales, D., Yablonsky, G. S., Marin, G. B. & Gleaves, J. T. Multi-zone TAP-reactors theory and application: I. The global transfer matrix equation. *Chem. Eng. Sci.* **56**, 133–149 (2001).
6. Miller, D. J. *et al.* Oxidation of Pt(111) under near-ambient conditions. *Phys. Rev. Lett.* **107**, 195502 (2011).
7. Porsgaard, S. *et al.* Stability of platinum nanoparticles supported on SiO₂/Si(111): a high-pressure X-ray photoelectron spectroscopy study. *ACS Nano* **6**, 10743–10749 (2012).
8. Wang, H. *et al.* Influence of size-induced oxidation state of platinum nanoparticles on selectivity and activity in catalytic methanol oxidation in the gas phase. *Nano Lett.* **13**, 2976–2979 (2013).
9. Meunier, F. C. *et al.* Synergy between metallic and oxidized Pt sites unravelled during room temperature CO oxidation on Pt/Ceria. *Angew. Chem. Int. Ed.* **60**, 3799–3805 (2021).

REVIEWER COMMENTS

Reviewer #1 (Remarks to the Author):

I agree with publication of this work in the present state

Reviewer #2 (Remarks to the Author):

I have well received the authors' rebuttal to my queries. I appreciate their efforts and value their consideration. I keep my initial view that the paper is very interesting and I add that it'd be very useful in future catalytic endeavors. However, risking being perceived as the now (in)famous reviewer#2 -which, according to my luck with the editorial system, I actually am!!!, I have to insist on the following issues:

1. Please, describe experimental techniques with sufficient detail. I agree that the manuscript is already a bit extensive, but experimental details can be placed in the SI where the interested reader can find them. The need for such detail in describing experiments cannot be further insisted upon when reproducibility issues are hotly debated in the literature (and even in the media!), please read: <https://nap.nationalacademies.org/catalog/25303/reproducibility-and-replicability-in-science>, <https://onlinelibrary.wiley.com/doi/full/10.1002/anie.201606591>, <https://www.annualreviews.org/doi/abs/10.1146/annurev-chembioeng-060718-030323>.

2. The following guidelines on reporting XPS measurements should be followed: <https://avs.scitation.org/doi/10.1116/1.5065501>, <https://avs.scitation.org/doi/10.1116/1.5140747>, <https://avs.scitation.org/doi/10.1116/6.0000661>, <https://avs.scitation.org/doi/10.1116/6.0000685>, <https://avs.scitation.org/doi/10.1116/6.0000377>.

3. The following guidelines on reporting physisorption experiments should be followed: <https://www.degruyter.com/document/doi/10.1515/pac-2014-1117/html?lang=en>. The experimental part should at least tell the following: (1) range of P/P0 used for tests, (2) number of data points recorded during the tests, (3) equilibration time, (4) procedure adopted for BET fitting -include CBET values in the report-, (5) Standard deviation of measurements. (6) Amounts of material for tests. (7) Outgassing procedure.

4. Concerning BET surface area, please include its value in the paper. According to the paper by Gleaves (App Catal A: Gen 160 (1997) 55) -that the authors cited in their rebuttal-: "Assuming a catalyst sample has a surface area of 10 m²/g, a single pulse would be equivalent to 1/1 000 000 of the total surface area of a 0.1 g sample. If the active surface area of the catalyst comprises a reasonable fraction (e.g., 20.1%) of the total surface area of the catalyst, then a single pulse will have a negligible effect on the active surface." Accordingly, I consider important to inform the reader whether the above assumption is fulfilled by the catalyst studied in the paper. I have to say that the reader needs to be informed of such details. By the way, formally, the SiO₂ support featured in the paper is non-nanoporous rather than "non-porous".

5. I thank the authors for suggesting the excellent papers by Gleaves & Constales et al. It was a very interesting and instructive read. However, after reading them, I don't know if I misread these papers, but

the statement: "As the primary transport mechanism is Knudsen diffusion any mass transfer effects can be minimised." made by the authors seems contradictory since the modelling of the TAP reactor considered by Gleaves actually includes mass transport effects. What the method claims is that diffusion constants can be discerned from apparent reaction constants after modelling. This then needs clarification.

6. Although TAP modelling was beautifully done by previous authors, it seems that some well established facts of mass transfer have not been considered so far. Particularly, the fact that "Bulk and Knudsen diffusion mechanisms occur together and it is prudent to take both mechanisms into account rather than assume that one or other mechanism is 'controlling'." -Krishna & Wesselingh, Chem Eng Sci 52 (1997) 861. Therefore, I recommend the authors being more prudent when making categorical statements about mass transport in the TAP reactor.

7. Concerning heat transport, the argument posed by the authors makes sense. But making sense does not constitute scientific proof. This particular issue is sensitive to the analysis of the TAP data because, once again according to Gleaves et al, the following assumptions are made when modelling the reactor: "The basis of the one-zone-model is the following [...] assumptions: [...] 2. There is no radial gradient of concentration in the catalyst bed. [...] 3. There is no axial or radial temperature gradient in the catalyst bed. [...]." Therefore, from the start, no heat transfer limitations are assumed -not proved!- for the experiments. Therefore, once again, I recommend being more prudent with the pen in the manuscript.

8. Concerning the analysis of XPS data, it is important to recall that 3 nm is considered to be the average depth for XPS made on a homogeneous sample. For multicomponent samples however, the depth of analysis depends on the particular elements of the sample and on the power at which the X-ray source is operated. Therefore, one may not directly conclude that the XPS data for a Pt/SiO₂ sample is "bulk". For prudence sake, one may rather say that XPS analysis is near-surface.

9. Concerning the results obtained with the SiO₂ support, "Minimal activity" is not the same as "zero activity". Therefore, again, prudence is advised. The conclusion should be toned down to say that most of the activity of the Pt/SiO₂ can be attributed to Pt neglecting its possible synergistic interactions with the SiO₂ support. There is plenty of evidence of the latter in the literature (e.g.

<https://www.sciencedirect.com/science/article/pii/S0021951722004675>;

<https://www.sciencedirect.com/science/article/pii/S0920586118310344> + others).

10. Concerning flux and concentration, I guess that flux is either the quantity of mass (or moles) per unit transport area which is (in a more general sense) the concentration of a given compound -or element- per unit area. Isn't it? Therefore, saying that "we are not measuring concentration of a gas stream, but instead measuring the flux of gas..." appears confusing. Therefore, I ask the authors to be a bit clearer about their definitions. Also, what is the "transport area" to which their flux is referred to?

11. Concerning the quantification via MS data, the problem is not with the noise but with the changes in baseline of the MS spectra. One may check this in the literature: e.g., J. Vac. Sci. Technol. A 5, 134–139 (1987), J. Am. Soc. Mass Spectrom. 2021, 32, 8, 2135–2143, etc. Therefore, once again a more careful language should be used in the paper.

12. Concerning the chemical state of Pt after reduction, XPS data dictates prudence, once again, because what one sees is that after H₂ treatment some fraction of Pt maybe remain oxidized. However, the low intensity of the presented Pt 4f (?) hinders interpretation. Also, I guess that authors meant "Partially spent Pt" in Figure SI-18.a

13. The justification for using the selected O₂-CO ratio of the work should be mentioned in the revised paper.

14. Concerning data fitting for estimating apparent activation energies, the response provided by the authors reaffirms the need to make covariance analysis of the data. This is due to the fact that conventional regression models are inherently based on normal distribution of data. It is suggested that the authors make a covariance analysis of the data:

https://en.wikipedia.org/wiki/Analysis_of_covariance

Design and analysis: A researcher's handbook (3rd ed.). Prentice-Hall, Inc.

I hope that the authors receive the above suggestions in a positive sense since I only do them in the spirit of helping improving the quality of the paper.

Reviewer #3 (Remarks to the Author):

Authors have revised the manuscript satisfactorily and all the points raised by the reviewer have been addressed. Now this manuscript can be accepted for publication.

Reviewer #4 (Remarks to the Author):

The authors have answered adequately to my earlier concerns and I believe that the present manuscript is sound and brings about a very interesting study and method to discriminate parallel reaction pathways on catalysts.

REVIEWER COMMENTS

Reviewer #1:

I agree with publication of this work in the present state.

Reviewer #3:

Authors have revised the manuscript satisfactorily and all the points raised by the reviewer have been addressed. Now this manuscript can be accepted for publication.

Reviewer #4:

The authors have answered adequately to my earlier concerns, and I believe that the present manuscript is sound and brings about a very interesting study and method to discriminate parallel reaction pathways on catalysts.

We would like to thank reviewers #1, #3, and #4 for their time and are glad that our changes to the manuscript were sufficient.

Reviewer #2:

I have well received the authors' rebuttal to my queries. I appreciate their efforts and value their consideration. I keep my initial view that the paper is very interesting and I add that it'd be very useful in future catalytic endeavors. However, risking being perceived as the now (in)famous reviewer#2 -which, according to my luck with the editorial system, I actually am!!!, I have to insist on the following issues:

1. Please, describe experimental techniques with sufficient detail. I agree that the manuscript is already a bit extensive, but experimental details can be placed in the SI where the interested reader can find them. The need for such detail in describing experiments cannot be further insisted upon when reproducibility issues are hotly debated in the literature (and even in the media!), please read: <https://nap.nationalacademies.org/catalog/25303/reproducibility-and-replicability-in-science>, <https://onlinelibrary.wiley.com/doi/full/10.1002/anie.201606591>, <https://www.annualreviews.org/doi/abs/10.1146/annurev-chembioeng-060718-030323>.

We completely agree that reproducibility is important in catalysis. However, in lieu of any specific suggestions from reviewer #2 we are confident that the methods and analysis are described sufficiently that a researcher in heterogeneous catalysis could reproduce the results.

2. The following guidelines on reporting XPS measurements should be followed: <https://avs.scitation.org/doi/10.1116/1.5065501>, <https://avs.scitation.org/doi/10.1116/1.5140747>, <https://avs.scitation.org/doi/10.1116/6.0000661>, <https://avs.scitation.org/doi/10.1116/6.0000685>, <https://avs.scitation.org/doi/10.1116/6.0000377>.

We have added the following statement to clarify the XPS sample preparation which missing in the revised manuscript:

The XPS sample pretreatments were performed in the DRIFTS reactor and resultant catalyst powder was pressed onto a carbon tape mounted on a Si wafer for each XPS measurement.”

All other reporting of the XPS data falls within the community guidelines.

3. The following guidelines on reporting physisorption experiments should be followed: <https://www.degruyter.com/document/doi/10.1515/pac-2014-1117/html?lang=en>. The experimental part should at least tell the following: (1) range of P/P0 used for tests, (2) number of data points recorded during the tests, (3) equilibration time, (4) procedure adopted for BET fitting -include CBET values in the report-, (5) Standard deviation of measurements. (6) Amounts of material for tests. (7) Outgassing procedure.

We have included the P/P0 range, the number of data points, equilibration time, standard deviation of measurements, amount of material for tests, model used for BET fitting, and the outgassing procedure in the figure caption so that it now reads:

“Supplementary Figure 13. N₂ adsorption isotherm plot for SiO₂ support material demonstrating non-nanoporous/macroporous behaviour. BET Measurements were performed using a micrometrics 3flex adsorption analyser. Pressure range: 0 – 1000 mbar, number of data points recorded: 62, equilibration interval: 10 to 45s, standard deviation of fit: 1.735 mmol/g. sample mass: 40 mg. BET Surface area: 514 m²/g. model: N2 - Tarazona NLDFT, Esf = 30.0K. The sample was degassed at 160 °C under vacuum (10⁻³ torr) for 600 min.”

4. Concerning BET surface area, please include its value in the paper. According to the paper by Gleaves (App Catal A: Gen 160 (1997) 55) -that the authors cited in their rebuttal-: "Assuming a catalyst sample has a surface area of 10 m²/g, a single pulse would be equivalent to 1/1 000 000 of the total surface area of a 0.1 g sample. If the active surface area of the catalyst comprises a reasonable fraction (e.g., 20.1%) of the total surface area of the catalyst, then a single pulse will have a negligible effect on the active surface." Accordingly, I consider important to inform the reader whether the above assumption is fulfilled by the catalyst studied in the paper. I have to say that the reader needs to be informed of such details.

The TAP experiment is considered to be state defining if a single pulse has negligible impact on the active surface. We direct the reviewer to Supplementary Figure 7a which demonstrates it takes ~400 pulses of reactant gas to saturate the surface in CO, and Figures 3a and 3b where it takes 200 pulse sets (600 pulses) to remove the adsorbed CO. While there is no specific definition of what constitutes as “state defining” we are confident this experiment fulfils that requirement and have added the line:

“Given the large number of pulses required to saturate/titrate the catalyst surface, the experiments fall well-within the “state defining” regime as required by the MZTRT model.”

We would like to state that the BET surface area is not relevant to the *total number of active sites* and therefore can be misleading and as such we made a conscious choice to not include the data. Further, we do not report any site-normalised activity data (e.g., turnover frequencies). However, the BET surface area of our support material is reported in the caption to Supplementary Figure 13.

By the way, formally, the SiO₂ support featured in the paper is non-nanoporous rather than "non-porous".

Reviewer #2 is correct, and we have re-worded non-porous to non-nanoporous

5. I thank the authors for suggesting the excellent papers by Gleaves & Constales et al. It was a very interesting and instructive read. However, after reading them, I don't know if I misread these papers, but the statement: "As the primary transport mechanism is Knudsen diffusion any mass transfer effects can be minimised." made by the authors seems contradictory since the modelling of the TAP reactor considered by Gleaves actually includes mass transport effects. What the method claims is that diffusion constants can be discerned from apparent reaction constants after modelling. This then needs clarification.

The primary mass transfer mechanism is Knudsen diffusion, which is precisely described by the 1D diffusion model. Differing from a conventional flow experiment at elevated conversions where the mass transfer is not precisely described and can adversely affect the calculation of kinetics, in the TAP 1D model the transport (mass transfer) is precisely described and the kinetics we calculate are not correlated to the mass transfer mechanism. As it seems that this line has caused some confusion, we have decided to remove it instead.

6. Although TAP modelling was beautifully done by previous authors, it seems that some well established facts of mass transfer have not been considered so far. Particularly, the fact that "Bulk and Knudsen diffusion mechanisms occur together and it is prudent to take both mechanisms into account rather than assume that one or other mechanism is 'controlling'" - Krishna & Wesselingh, Chem Eng Sci 52 (1997) 861. Therefore, I recommend the authors being more prudent when making categorical statements about mass transport in the TAP reactor.

While increasingly complex models may model the physics of a TAP reactor under a wider range of boundary conditions, the model currently employed captures all the physics relevant to the TAP experiment as we apply it as we precisely recreate the transient behaviour of our inert tracer (see Figure 4b). It is not the purview of this paper to precisely describe the physics (and the limitations) of the TAP experiment which has been described extensively in the previous literature that is cited in this work. As such, we feel a reference to the many papers where this has been discussed is more than sufficient. If the author would like a full description of how non-ideal conditions effect TAP reactor systems, we direct them to the fantastic work of Constales (<https://doi.org/10.1016/j.ces.2005.10.022>), where they empirically demonstrate that the 1D model is valid for a wide range of conditions, which our experiment falls well within, please see our previous publication on the TAP reactor design utilised in this work where we demonstrate that the 1D model is indeed fulfilled (<https://doi.org/10.1016/j.ces.2023.147489>).

7. Concerning heat transport, the argument posed by the authors makes sense. But making sense does not constitute scientific proof. This particular issue is sensitive to the analysis of the TAP data because, once again according to Gleaves et al, the following assumptions are made when modelling the reactor: "The basis of the one-zone-model is the following [...] assumptions: [...] 2. There is no radial gradient of concentration in the catalyst bed. [...] 3. There is no axial or radial temperature gradient in the catalyst bed. [...]" Therefore, from the start, no heat transfer limitations are assumed -not proved!- for the experiments. Therefore, once again, I recommend being more prudent with the pen in the manuscript.

We measure the temperature of our catalyst using a thermocouple inserted into the catalyst bed, which is very thin, meaning that we are confident in the accuracy of our temperature measurement of our catalyst. Given the highly complex nature of the TAP reactor, and small size of the microreactor housing the catalyst, it is not physically possible to insert multiple

thermocouples and/or a calorimeter inside our catalyst bed to confirm that there are no heat thermal gradients. However, we would like to direct reviewer #2 to Figure 4b. Our transport curves are reproduced precisely by the thin-zone (an extension of the one-zone) model. If there were any **significant** temperature gradients observed in the experiment, by definition the model would not apply, which is not the case here. It is not feasible to remove all temperature gradients in an experiment but given that we are able to precisely describe the transport using the 1D model, and that we measure the temperature of our catalyst directly, we are confident in its validity. While making sense does not constitute a scientific proof, in the absence of idealised system, it is more than sufficient.

8. Concerning the analysis of XPS data, it is important to recall that 3 nm is considered to be the average depth for XPS made on a homogeneous sample. For multicomponent samples however, the depth of analysis depends on the particular elements of the sample and on the power at which the X-ray source is operated. Therefore, one may not directly conclude that the XPS data for a Pt/SiO₂ sample is "bulk". For prudence sake, one may rather say that XPS analysis is near-surface.

We would like to clarify that we do not mention that the XPS measurement is a "bulk" measurement in the manuscript but state in the revision comments:

"We consider the XPS measurement to be a bulk chemical analysis for the 2 nm Pt nanoparticles because the XPS probing depth is ~3 nm using an Al K α X-ray source."

We want to clarify in the statement that a probing depth of ~3nm does indeed probe the bulk region of 2 nm Pt nanoparticles (~1 nm depth). It is well established that the mean free path of electrons slightly changes through different elemental samples (<https://doi.org/10.1002/sia.740010103>) but this does not affect our original assessment (<https://doi.org/10.1002/sia.5789>).

9. Concerning the results obtained with the SiO₂ support, "Minimal activity" is not the same as "zero activity". Therefore, again, prudence is advised. The conclusion should be toned down to say that most of the activity of the Pt/SiO₂ can be attributed to Pt neglecting its possible synergistic interactions with the SiO₂ support. There is plenty of evidence of the latter in the literature (e.g. <https://www.sciencedirect.com/science/article/pii/S0021951722004675>; <https://www.sciencedirect.com/science/article/pii/S0920586118310344> + others).

The first paper is discussing the role that SiO₂ plays in hydrogen spillover, which is not relevant to CO oxidation. The second paper talks about the role of synergistic physical mixtures of catalytically active materials – not catalysts and inert supports. We are very confident in the inert nature of our SiO₂ support material, particularly for the TPO experiments which constitute the core part of this work, we have however reworded the sentence so that it is clearer now reads:

"Minimal activity was recorded over the SiO₂ (blank) sample (Supplementary Fig. 17) during co-pulsed experiments, with zero activity recorded during a TPO experiment, meaning the catalytic activity recorded during the TPO experiments is solely prescribed to the Pt nanoparticles."

10. Concerning flux and concentration, I guess that flux is either the quantity of mass (or moles) per unit transport area which is (in a more general sense) the concentration of a given compound -or element- per unit area. Isn't it? Therefore, saying that "we are not measuring concentration of a gas stream, but instead measuring the flux of gas..." appears confusing. Therefore, I ask the

authors to be a bit clearer about their definitions. Also, what is the "transport area" to which their flux is referred to?

The exit flux is the number of molecules leaving the TAP reactor per unit area per second of the microreactor exit, which is 0.2 cm.

11. Concerning the quantification via MS data, the problem is not with the noise but with the changes in baseline of the MS spectra. One may check this in the literature: e.g., J. Vac. Sci. Technol. A 5, 134–139 (1987), J. Am. Soc. Mass Spectrom. 2021, 32, 8, 2135–2143, etc. Therefore, once again a more careful language should be used in the paper.

We are well aware that mass spectrometry is not a 1:1 relationship between concentration and explicitly describe how we perform our quantification in the supplementary information. In lieu of any specific language being pinpointed by the reviewer, we are happy with our discussion on how we perform our quantitative mass spec.

12. Concerning the chemical state of Pt after reduction, XPS data dictates prudence, once again, because what one sees is that after H₂ treatment some fraction of Pt maybe remain oxidized. However, the low intensity of the presented Pt 4f (?) hinders interpretation.

There is no conclusive evidence of a partially oxidized Pt species in our XPS data after a H₂ reduction treatment considering the signal to noise ratio (SNR) and peak full width at half maxima (FWHM). We would like to clarify that the relatively high SNR is due to a Pt weight loading of 0.72% which is close to the detection limit of conventional XPS lab-based systems. The implication of this is that an appropriate pass energy (which controls the FWHM) must be set on the detector which allows for sufficient intensity to detect species. Further adjustment of the pass energy to decrease the FWHM in order to be able to determine if this is a minor contribution of partially optimized Pt species is not feasible with our Pt/SiO₂ sample on conventional lab based XPS systems. we cannot definitively rule out the presence of partially oxidized Pt by XPS but refer the reviewer to our previous DRIFTS evidence for solely metallic Pt identified in our previous response. We would caution reviewer #2 to not overanalyse noise, which is inherent in every measurement.

With regards to: "the presented Pt 4f (?)". The caption states that the XPS spectra is of the Pt 4f region.

Also, I guess that authors meant "Partially spent Pt" in Figure SI-18.

We can see no errors with Figure SI-18.

13. The justification for using the selected O₂-CO ratio of the work should be mentioned in the revised paper.

We have added the sentence:

"The 1:2 ratio of CO:O₂ was chosen to allow excess O₂ in the reactant stream to act as a titrant to sequentially remove the preadsorbed CO* to probe the catalytically activity as a function of CO* coverage."

14. Concerning data fitting for estimating apparent activation energies, the response provided by the authors reaffirms the need to make covariance analysis of the data. This is due to the fact that

conventional regression models are inherently based on normal distribution of data. It is suggested that the authors make a covariance analysis of the data: https://en.wikipedia.org/wiki/Analysis_of_covariance Design and analysis: A researcher's handbook (3rd ed.). Prentice-Hall, Inc.

This level of analysis is unnecessary. The linear regression of apparent activation energies is a well-established protocol and is not specific enough to require this level of analysis. Standard line-fitting programs as present in Origin, Excel, MATLAB, or LabVIEW is more than sufficient for this dataset.

We reference the reviewer to recent publications in Nature Communications which demonstrate our analysis is consistent with standard procedures in heterogeneous catalysis research:

<https://doi.org/10.1038/s41467-022-28366-w>

<https://doi.org/10.1038/s41467-023-36339-w>

<https://doi.org/10.1038/s41467-021-22946-y>

I hope that the authors receive the above suggestions in a positive sense since I only do them in the spirit of helping improving the quality of the paper.

REVIEWERS' COMMENTS

Reviewer #2 (Remarks to the Author):

Dear authors,

Provided the nature and tone of your responses, I have nothing further to comment about the paper.

REVIEWER COMMENTS

Reviewer #2:

Dear authors,

Provided the nature and tone of your responses, I have nothing further to comment about the paper.

We recognise that it is not easy to perform a robust and rigorous peer review. The feedback has significantly helped with readership and providing the required information for others to reproduce this work. As such, we would like to thank reviewer #2 for their extensive effort in reviewing this paper.